# LARFT: Closing the Cognition-Action Gap for Length Instruction Following in Large Language Models

Wei Zhang [* 1]   Lintong Du [* 1]   Yuanhe Zhang [1]   Zhenhong Zhou [2]   Kun Wang [2]   Li Sun [1]   Sen Su [1 3]

## Abstract

Despite the strong performance of Large Language Models (LLMs) on complex instruction-following tasks, precise control of output length remains a persistent challenge. Existing methods primarily attempt to enforce length constraints by externally imposing length signals or optimization objectives, while largely overlooking the underlying limitation: the model's intrinsic deficit in length cognition. To address this, we propose **LARFT** (**L**ength-**A**ware **R**einforcement **F**ine-**T**uning), a training framework that aligns the model's length cognition with its action. Specifically, LARFT integrates length-oriented reinforcement learning with a hindsight length awareness. By transforming on-policy data into hindsight self-awareness tasks where the model learns to identify the actual length of its own generation, LARFT jointly optimizes the model's internal representation of length information and refines its policy to satisfy length constraints, thereby achieving precise and reliable length instruction following. Extensive experiments across four base models demonstrate that LARFT outperforms existing baselines, achieving an average improvement of **+20.92** points across three length instruction following benchmarks with only a marginal decline of **-1.45** points on four general capability benchmarks. Our code is available at https://github.com/Captain-zhangw/LARFT.

## 1. Introduction

Large Language Models (LLMs) have demonstrated remarkable capabilities in accurately accomplishing various complex instructions (Ouyang et al., 2022). Nevertheless, following precise length constraints remains a distinct and persistent challenge (Yuan et al., 2025b). Beyond the complexity of the task itself, the token-based nature of tokenization (Wang et al., 2020) and the intrinsic limitations of model architectures (Brown et al., 2020) impede the comprehension of real-world length concepts (e.g., "100 words"), causing models to struggle significantly with length instruction following. This issue becomes increasingly critical with the expansion of context windows and the growing demand for long-form generation (Peng et al., 2023; Bai et al., 2024). Existing models fail to scale their length-following capabilities synchronously with the rapid expansion of model output windows (Zhang et al., 2025a). Consequently, in real-world applications such as creative writing and report generation, models struggle to align with precise length constraints, producing outputs that either fall significantly short or become excessively verbose (Que et al., 2024; Zhang et al., 2025b).

Prior efforts to enhance length instruction following capabilities typically fall into two paradigms: external length marker incorporation and length-constrained policy optimization. The former, utilizing techniques like length-specific tokens (Li et al., 2024; Yuan et al., 2025a), relies on rigid and intrusive interventions to mechanically enforce length limits. Consequently, these methods often incur additional inference overhead and struggle to generalize to long-form generation. The latter treats length constraints as reward signals, utilizing policy optimization for direct alignment (Jie et al., 2024; Aggarwal & Welleck, 2025). However, given that policy updates are driven by dense token-level semantic dependencies, this approach inevitably entangles length constraints with semantic features in the representation space. This lack of separation makes it difficult to control length independently of content, leading to low sample efficiency and imprecise control. Fundamentally, we attribute these limitations to a missing cognitive basis: the absence of an internal representation of length (Zhang et al., 2025a). Without establishing a distinct latent concept of length during generation, training models to follow

---

*Equal contribution   [1]Beijing University of Posts and Telecommunications, Beijing, China   [2]Nanyang Technological University, Singapore, Singapore   [3]Chongqing University of Posts and Telecommunications.  Correspondence to: Sen Su <susen@bupt.edu.cn>.

*Proceedings of the 43$^{rd}$ International Conference on Machine Learning*, Seoul, South Korea. PMLR 306, 2026. Copyright 2026 by the author(s).

such constraints becomes an ill-posed optimization problem. Consequently, the models fail to effectively decouple length from semantics, leading to suboptimal convergence and a persistent inability to meet strict length constraints.

Building upon this insight, we propose **LARFT** (**L**ength-**A**ware **R**einforcement **F**ine-**T**uning), a unified training framework designed to close the cognition-action gap in length instruction following. As shown in Figure 1, LARFT integrates three core components: **(I) Length-Oriented Reinforcement Learning**: We employ GRPO (Shao et al., 2024) with a verifiable length reward to explicitly guide the model's generation action towards the target constraint. **(II) Hindsight Length Awareness**: We introduce a mechanism that transforms length-mismatched generation trajectories into valuable supervision. By relabeling the model's own outputs with a cognition-inducing prompt (e.g., *"Count the words in the text above"*), we train the model to count the length of its generated content. This allows the model to internalize the concept of length from its own generations, re-purposing on-policy generations as valid instances for length cognition. **(III) Unified Optimization Mechanism**: We employ a joint optimization strategy that dynamically balances length awareness and instruction following objectives. This mechanism prioritizes establishing length cognition in the early training stages and progressively shifts focus to precise generation control. By integrating these three components, LARFT fosters a positive feedback loop where enhanced awareness and generation capabilities mutually reinforce, leading to robust length instruction following.

We empirically evaluate LARFT across four different models on three length instruction following benchmarks and four general capability benchmarks. The results demonstrate that LARFT achieves state-of-the-art performance, improving length-following capability by an average of **20.92** points compared to base models and surpassing the strongest RL baseline by **4.59** points. Crucially, this specialized enhancement incurs negligible cost to general performance: we observe a slight improvement in generation quality (**+2.22** points) and only a minor dip in general capabilities (**-1.45** points). Through ablation and interpretability analyses, we verify the robustness of our method and its role in facilitating length awareness and following.

Our main contributions are summarized as follows:

- We identify the lack of intrinsic length cognition as a key bottleneck in length instruction following and propose the **LARFT** framework to align the model's length cognition with its generation action.

- We demonstrate the effectiveness of LARFT across various models. On length instruction following benchmarks, LARFT achieves an average improvement of **20.60** points over untrained models and **4.59** points over the second-

best method.

- We validate the critical role and robustness of the length awareness and provide an interpretability analysis explaining how training for length cognition explicitly facilitates more accurate generation.

## 2. Related Works

**Length Instruction Following Capabilities of LLMs.** The ability of LLMs to follow length instructions is constrained by their autoregressive nature and positional encoding limitations (Holtzman et al., 2020; Kazemnejad et al., 2023; Butcher et al., 2025). Existing methods to improve length controllability can be broadly categorized into three streams. Supervised fine-tuning approaches attempt to align output content with length constraints by constructing length-specific tokens or prompts during training (Wang et al., 2024; Li et al., 2024; Song et al., 2025). Reinforcement learning approaches model length constraints as reward functions, guiding the model towards desired lengths through policy optimization (Yuan et al., 2025b; Jie et al., 2024; Li et al., 2025; Singhal et al., 2023). Meanwhile, inference-time interventions adjust the decoding process, often by inserting placeholder tokens to meet target lengths (Xiao et al., 2026; Yuan et al., 2025a; Gu et al., 2025; Akinfaderin et al., 2025). Nevertheless, viewing length as an external constraint rather than an internal semantic feature limits their ability to achieve precise control.

**Long-form Generation of LLMs.** While the context windows of LLMs have significantly expanded (Kamradt, 2023; Liu et al., 2024b), the capability for long-form generation does not linearly scale with input capacity (Bai et al., 2024). The emergence of reasoning models demonstrates the utility of extended generation, where prolonged reasoning processes enhance performance on complex mathematical and reasoning tasks (Snell et al., 2024; Guo et al., 2025). However, such unconstrained generation inevitably incurs excessive token consumption and increased inference latency (Aggarwal & Welleck, 2025; Sui et al., 2025). Consequently, precise control over output length is imperative to balance resource efficiency with generation quality, particularly for applications requiring specific constraints, such as creative writing and question answering (Quan et al., 2024; Que & Rong, 2025).

## 3. Method

In this section, we detail **Length-Aware Reinforcement Fine-Tuning (LARFT)**, a unified framework designed to close the cognition-action gap in length instruction following. LARFT synergizes three core components: **Length-Oriented Reinforcement Learning (§3.1)** optimizes the

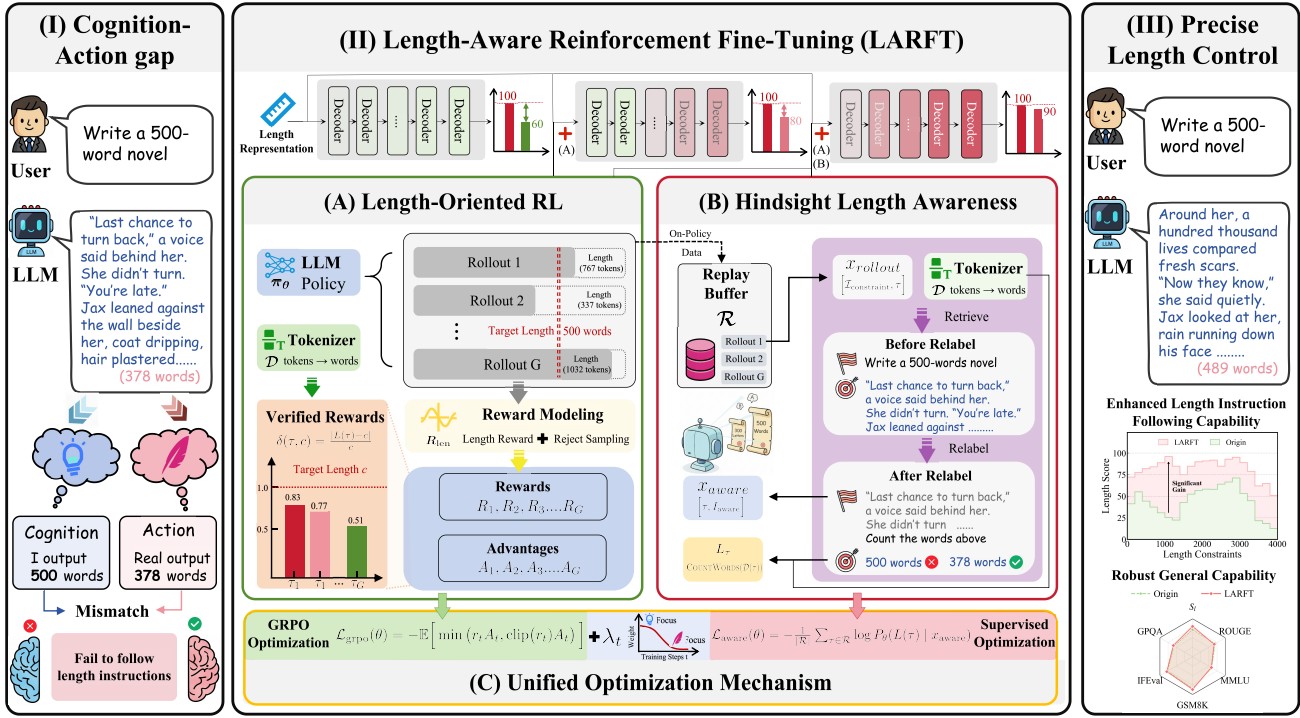

*Figure 1.* Overview of the Length-Aware Reinforcement Fine-Tuning (LARFT) framework. (I) Illustrates the Cognition-Action Gap, where standard LLMs fail to align their output length with user instructions. (II) Presents our LARFT method, which unifies Length-Oriented RL and Hindsight Length Awareness to align generation with constraints. (III) Demonstrates that LARFT achieves precise length control while maintaining robust general capabilities.

generation policy via explicit reward signals, while **Hindsight Length Awareness (§3.2)** fosters internal length concepts by repurposing on-policy trajectories. These components are integrated via a **Unified Optimization Mechanism (§3.3)**, which dynamically balances the training objectives to ensure the co-evolution of the model's awareness and generation capabilities.

### 3.1. Length-Oriented Reinforcement Learning

The objective of length instruction following naturally aligns with the paradigm of *Reinforcement Learning with Verifiable Rewards* (RLVR). As a quantifiable metric, text length can be directly formulated as a verifiable reward signal. Consequently, we employ *Group Relative Policy Optimization* (GRPO), a highly efficient policy gradient algorithm, to explicitly optimize the model's length-following capabilities.

#### 3.1.1. VERIFIED LENGTH REWARD FORMULATION

To effectively guide the model towards precise constraints, we need to transform the discrete word counts into a dense reward signal that reflects the deviation from the target. Formally, let $\mathcal{S}$ denote the space of natural language strings and $\mathcal{V}$ denote the vocabulary of tokens. We consider a raw instruction $\mathcal{I} \in \mathcal{S}$ which explicitly specifies a target length constraint $c$ (measured in words). The instruction is processed into a model input prompt $x$. Let $\tau = (o_1, o_2, \ldots, o_T)$ denote a candidate response sequence, where $o_t \in \mathcal{V}$ represents the token generated at time step $t$. To calculate the word count from the token sequence $\tau$, we define the length measurement function $L : \mathcal{V}^* \to \mathbb{N}$ based on the inverse mapping (detokenizer) $\mathcal{D} : \mathcal{V}^* \to \mathcal{S}$. Formally, the length is given by:

$$L(\tau) = \text{COUNTWORDS}(\mathcal{D}(\tau)), \quad (1)$$

where $\text{COUNTWORDS}(\cdot)$ calculates the number of words in a string. Based on this metric, we first quantify the normalized absolute deviation $\delta$ of the actual length $L(\tau)$ relative to the target $c$:

$$\delta(\tau, c) = \frac{|L(\tau) - c|}{c}. \quad (2)$$

Using $\delta$, we define the length-specific reward function $R_{\text{len}}(\tau, c)$ as a piecewise linear function:

$$R_{\text{len}}(\tau, c) = \max\left(0, 1 - \delta(\tau, c)\right). \quad (3)$$

This formulation yields a bounded reward value in $[0, 1]$, providing a stable and dense signal for the advantage estimation in the subsequent optimization process.

### 3.1.2. GROUP RELATIVE POLICY OPTIMIZATION

We leverage GRPO to explicitly align the model with the verifiable length reward $R_{\text{len}}$. Rather than relying on a learned value function, GRPO utilizes group-based rollouts to estimate the advantage. Specifically, for each prompt $x$ with a target length constraint $c$, we sample a group of $G$ valid trajectories $\{\tau_1, \ldots, \tau_G\}$ from the current policy $\pi_{\theta_{\text{old}}}$, where we explicitly discard degenerate sequences characterized by excessive repetition (e.g., n-gram loops). Using the group statistics as the baseline, we compute the advantage for each trajectory $\tau_i$ by normalizing the rewards:

$$A_i = \frac{R_{\text{len}}(\tau_i, c) - \mu_R}{\sigma_R}, \qquad (4)$$

where $\mu_R$ and $\sigma_R$ denote the mean and standard deviation of the rewards within the sampled group.

The policy is then updated by maximizing a clipped surrogate objective. Let $o_{i,t}$ denote the $t$-th token of the $i$-th trajectory $\tau_i$. We define the probability ratio $r_{i,t}(\theta) = \frac{\pi_\theta(o_{i,t}|o_{i,<t},x)}{\pi_{\theta_{\text{old}}}(o_{i,t}|o_{i,<t},x)}$, the optimization objective is defined as:

$$\mathcal{L}_{\text{GRPO}}(\theta) = -\frac{1}{\sum_{i=1}^{G}|\tau_i|}\sum_{i=1}^{G}\sum_{t=1}^{|\tau_i|} \min\Big[r_{i,t}(\theta)A_i,$$
$$\text{clip}\big(r_{i,t}(\theta); 1-\epsilon, 1+\epsilon\big)A_i\Big], \quad (5)$$

where the objective averages over the sampled group and sums across all tokens in each trajectory $\tau_i$, with $\epsilon$ denoting the clipping hyperparameter.

### 3.2. Hindsight Length Awareness

While length-oriented reinforcement learning offers a clear objective, direct optimization faces two primary hurdles: (i) Difficulty in Constraint Alignment: Since optimization applies to the entire sequence during RL updates, the gradient updates for length following are inevitably entangled with semantic objectives. (ii) Sample Inefficiency in Updates: A sample with a low length reward may still yield a high advantage if it outperforms its peers within a poor-quality group in GRPO. To address these challenges, we introduce the **Hindsight Length Awareness** mechanism. Inspired by Hindsight Experience Replay (Andrychowicz et al., 2017), which transforms failed trials into valid experiences, we aim to convert trajectories sampled during RL updates into more effective supervisory signals targeting length capabilities. This allows us to decouple length signals from semantic information, leveraging every on-policy sample as a valid instance of its own length for dual optimization. The implementation of this mechanism unfolds in three stages:

**On-Policy Data Sampling.** To recycle the diverse length samples produced during rollout, we maintain a replay buffer $\mathcal{R}$. Following the rollout phase, we not only compute policy gradients based on these trajectories but also persist them in $\mathcal{R}$. Since these trajectories originate from the model's current policy distribution, they accurately capture its intrinsic biases regarding length constraints. Viewing them as structurally aligned with the model's current capabilities, we retain these experiences for subsequent reconstruction into successful examples.

**Hindsight Awareness Relabeling.** Given a sample $(x, \tau)$ retrieved from the replay buffer $\mathcal{R}$, where $x$ is the input and $\tau$ is the generated output, the primary challenge lies in utilizing it for effective supervision. Since these trajectories exhibit an inherent entanglement of semantic content and length characteristics, direct goal relabeling (e.g., updating the length constraint $c$ in $x$ to the actual length of $\tau$) fails to mitigate the influence of semantics. To address this, we incorporate an awareness instruction, specifically *"Count how many words are in the text above?"*, to guide the model to self-evaluate the length of its output. We construct a composite input by concatenating the generated output $\tau$ with the awareness instruction $\mathcal{I}_{\text{aware}}$. Formally, the reformulated input $x_{\text{aware}}$ is defined as:

$$x_{\text{aware}} = [\tau; \mathcal{I}_{\text{aware}}], \qquad (6)$$

where $[\cdot; \cdot]$ denotes the concatenation operation. This reformulation shifts the learning objective from semantic generation to length awareness, effectively disentangling length characteristics from semantic dependencies by redirecting the model's attention exclusively to length representation.

**Supervised Optimization.** Having effectively decoupled the length constraint from semantic content via relabeling, we leverage these processed trajectories to construct a new length-awareness task. Specifically, we treat trajectories that fail the original task as valid training examples for a simpler objective: learning to be aware of its own output length. Moreover, unlike the original task that requires extensive exploration, the ground truth for the awareness task is deterministic and intrinsic. The correct label can be easily derived by measuring the length of the trajectory. This characteristic allows us to convert a high-variance exploration problem into a stable supervised optimization objective. We pair the reformulated input $x_{\text{aware}}$ with the ground-truth label $L(\tau)$, which represents the actual length of the trajectory $\tau$. We define the awareness loss $\mathcal{L}_{\text{aware}}$ as the negative log-likelihood of predicting the actual length given the model's own generation:

$$\mathcal{L}_{\text{aware}}(\theta) = -\frac{1}{|\mathcal{R}|}\sum_{\tau \in \mathcal{R}} \log P_\theta(L(\tau) \mid x_{\text{aware}}). \qquad (7)$$

This objective drives the model to retrospectively analyze its output, thereby inducing an internal representation of length. This mechanism effectively closes the loop between cognition and action, fostering a "Hindsight Length Awareness" cycle that iteratively reinforces length instruction following capabilities.

### 3.3. Unified Optimization Mechanism

To integrate the proposed strategies, we formulate a joint training objective that dynamically balances reward maximization with length grounding. At training step $t$, the unified loss $\mathcal{L}_{\text{unified}}^{(t)}$ and the dynamic weighting coefficient $\lambda_t$ are defined as:

$$\mathcal{L}_{\text{unified}}^{(t)}(\theta) = \mathcal{L}_{\text{GRPO}}(\theta) + \lambda_t \cdot \mathcal{L}_{\text{aware}}(\theta),$$
$$\lambda_t = \frac{\lambda_{\max}}{2}\left[1 + \cos\left(\frac{t\pi}{T}\right)\right], \quad (8)$$

where $\mathcal{L}_{\text{GRPO}}$ is the policy optimization objective (Eq. 5), $\mathcal{L}_{\text{aware}}$ is the awareness loss (Eq. 7), $T$ denotes the total training steps, and $\lambda_{\max}$ is the initial weight. We employ a Cosine Annealing schedule for $\lambda_t$ to establish a coarse-to-fine learning curriculum: the model prioritizes structural length alignment in the early stages ($\lambda_t \approx \lambda_{\max}$) and progressively shifts focus toward semantic instruction following and reward maximization as training proceeds ($\lambda_t \to 0$).

In summary, LARFT effectively closes the feedback loop between cognition and action. This holistic approach transcends mere surface-level instruction following, aligning the model's internal understanding of length with its external execution—effectively bridging the gap between "knowing" and "doing". The complete training procedure is formally detailed in Appendix A

## 4. Experimental Setup

### 4.1. Training Datasets

We establish a data construction pipeline to curate a large-scale length instruction following dataset. We source our raw data from AM-DeepSeek-R1-Distilled-1.4M (Zhao et al., 2025) and Chinese-DeepSeek-R1-Distill-data-110k-SFT (Cong et al., 2025) which distilled from DeepSeek-R1 (Guo et al., 2025) in English and Chinese respectively. These datasets cover a wide range of task categories, including instruction following, creative writing, QA, and reasoning, among others. We implement a rigorous data construction pipeline to filter out incompatible tasks and synthesize length-constrained instructions. After this curation process, the final dataset comprises 8,732 samples, each paired with a specific length constraint ranging from 10 to 4,000 words. Detailed descriptions of the data construction pipeline are provided in Appendix B.

### 4.2. Evaluation

We evaluate our method on three widely used length instruction following benchmarks. For comprehensive evaluation, we employ LIFEBench (Zhang et al., 2025a), which covers a wide range of length constraints. We report the Length Score (LS) and Length Deviation (LD) as primary metrics. For short-form constraints, we evaluate on Lenctrl-Bench (Wang et al., 2024), reporting MAE and ROUGE-L to measure length precision and generation quality, respectively. For long-form constraints, we use LongBench (Bai et al., 2024) and report $S_l$ and $S_q$ to assess length following and response quality. Additionally, we evaluate our model across four general benchmarks, including MMLU (Hendrycks et al., 2021), GSM8k (Cobbe et al., 2021), IFEval (Zhou et al., 2023), and GPQA (Rein et al., 2024), to ensure that optimizing for length does not compromise general capabilities. For all evaluations, the decoding temperature is set to 0.6. We restrict our evaluation to targets under 4,000 words due to the maximum output length limitations of the base models. Detailed settings are provided in Appendix C.

### 4.3. Baseline Methods

We compare LARFT with a comprehensive set of baselines, categorized into three groups based on their training strategies: (i) *Untuned Baseline:* The original instruct model without any additional tuning. (ii) *General Training Paradigms:* Standard alignment methods applied to our datasets, including SFT (supervised fine-tuning), RL (optimized directly via GRPO), and SFT+RL (the sequential SFT-then-RL paradigm). (iii) *Tailored Length-Following Methods:* Techniques specifically tailored for length instruction following tasks, specifically *Ruler* (Li et al., 2024), which utilizes meta-tokens to represent distinct length intervals, and *PositionID* (Wang et al., 2024), which explicitly incorporates word counts during training data construction. We conduct experiments on four representative open-source models: Qwen2.5-3B, Qwen2.5-7B (Yang et al., 2024), Llama-3.2-3B, and Llama-3.1-8B (Grattafiori et al., 2024).

### 4.4. Implementation Details

For SFT experiments, we set the global batch size to 64 and train for 3 epochs. For RL training, we include a KL penalty with a coefficient of $\beta = 0.001$. The rollout batch size is set to 128, while the update batch size is 32. We generate 4 rollouts per prompt and train for 3 epochs. Regarding LARFT, we set the hyperparameter $\lambda_{\max}$ to 0.01. All training runs are conducted on a cluster with $8\times$ NVIDIA A100 80GB GPUs. We provide comprehensive hyperparameter configurations in Appendix E.

*Table 1.* Main results on length-following tasks and general benchmarks. Bold values indicate the best performance.

| Method | In-Distribution Tasks (Length Following) | | | | | | Out-of-Distribution Tasks (General) | | | |
| --- | --- | --- | --- | --- | --- | --- | --- | --- | --- | --- |
| | LIFEBench | | LongBench | | Lenctrl-Bench | | MMLU ↑ | GSM8K ↑ | IFEval ↑ | GPQA ↑ |
| | LD ↓ | LS ↑ | $S_l$ ↑ | $S_q$ ↑ | MAE ↓ | ROUGE-L ↑ | | | | |
| *Qwen2.5-3B-Instruct* | | | | | | | | | | |
| Origin | 38.79 | 46.04 | 76.38 | 68.49 | 28.76 | 18.08 | 65.07 | **76.19** | 73.02 | **31.03** |
| SFT | 65.60 | 26.93 | 79.88 | 66.04 | 45.44 | 16.87 | 65.00 | 76.04 | 66.79 | 29.24 |
| RL | 30.71 | 54.10 | 87.23 | 69.49 | 13.92 | **20.86** | 65.00 | 74.83 | **74.10** | 30.13 |
| SFT+RL | 32.48 | 52.23 | 79.46 | 66.49 | 12.09 | 20.01 | 54.75 | 74.15 | 72.18 | 29.46 |
| PositionID | 64.95 | 27.28 | 74.07 | 63.11 | 84.9 | 19.17 | 59.97 | 72.86 | 56.38 | 28.35 |
| Ruler | 62.15 | 28.85 | 77.00 | 63.73 | 79.8 | 19.78 | 41.94 | 75.82 | 37.71 | 27.90 |
| LARFT | **19.96** | **66.40** | **88.34** | **73.61** | **7.72** | 20.25 | **66.46** | 74.22 | 71.34 | 29.01 |
| *Qwen2.5-7B-Instruct* | | | | | | | | | | |
| Origin | 30.96 | 53.84 | 78.89 | 78.55 | 24.01 | 19.90 | **73.59** | 83.70 | 80.58 | 33.04 |
| SFT | 60.12 | 30.05 | 82.04 | 80.83 | 47.77 | 17.85 | 73.48 | **84.91** | 77.10 | 32.14 |
| RL | 17.17 | 70.94 | 93.84 | 75.72 | 9.01 | **21.38** | 73.13 | 84.69 | 79.74 | 32.59 |
| SFT+RL | 21.42 | 65.15 | 79.46 | **86.07** | 9.71 | 19.80 | 72.67 | 84.69 | 73.86 | 32.14 |
| PositionID | 57.67 | 31.56 | 83.14 | 77.59 | 53.0 | 19.45 | 72.04 | 87.19 | 66.17 | 33.26 |
| Ruler | 57.22 | 31.84 | 83.77 | 76.72 | 65.4 | 19.40 | 71.02 | 84.23 | 48.24 | 31.70 |
| LARFT | **10.80** | **80.57** | **96.75** | 82.09 | **7.00** | 21.12 | 72.94 | 84.61 | **81.53** | **33.48** |
| *Llama-3.2-3B-Instruct* | | | | | | | | | | |
| Origin | 44.66 | 40.93 | 57.73 | 48.90 | 11.83 | 21.12 | 61.50 | **77.26** | **81.18** | **32.37** |
| SFT | 61.88 | 29.01 | 73.09 | 47.36 | 48.16 | 17.54 | **61.74** | 73.92 | 70.62 | 31.25 |
| RL | 22.30 | 64.02 | 88.25 | 48.23 | 5.49 | **21.73** | 61.35 | 76.04 | 78.42 | 31.47 |
| SFT+RL | 20.41 | 66.48 | 83.88 | 54.17 | 8.52 | 20.65 | 60.68 | 71.50 | 71.94 | 29.91 |
| PositionID | 63.57 | 28.04 | 72.35 | 51.49 | 72.2 | 19.40 | 23.91 | 69.52 | 60.26 | 24.33 |
| Ruler | 62.11 | 28.87 | 79.35 | 51.68 | 86.0 | 19.43 | 24.05 | 66.87 | 39.74 | 24.33 |
| LARFT | **17.95** | **69.84** | **88.71** | **54.78** | **4.99** | 21.22 | 60.87 | 74.83 | 79.14 | 30.58 |
| *Llama-3.1-8B-Instruct* | | | | | | | | | | |
| Origin | 41.13 | 43.93 | 64.75 | 62.90 | 12.74 | 21.59 | **68.45** | 82.49 | **85.85** | 31.25 |
| SFT | 44.96 | 42.64 | 80.87 | **64.84** | 54.40 | 17.02 | 67.73 | **84.00** | 76.27 | **32.81** |
| RL | 19.74 | 67.39 | 90.38 | 58.88 | **6.57** | 21.70 | 66.87 | 74.15 | 73.38 | 32.14 |
| SFT+RL | 21.43 | 65.14 | 71.26 | 63.43 | 14.19 | 17.91 | 65.05 | 77.48 | 64.75 | 28.35 |
| PositionID | 53.95 | 33.99 | 79.15 | 66.13 | 59.20 | 19.17 | 40.35 | 81.12 | 63.22 | 25.45 |
| Ruler | 54.65 | 33.52 | 79.35 | 64.47 | 73.8 | 19.66 | 37.15 | 79.08 | 48.61 | 24.33 |
| LARFT | **12.73** | **77.52** | **94.75** | 62.41 | 6.98 | **21.78** | 67.30 | 74.52 | 81.35 | 31.25 |

# 5. Experiments

## 5.1. Main Results

We present the comprehensive evaluation results in Table 1. LARFT establishes new state-of-the-art performance in length instruction following while preserving general capabilities, significantly outperforming other baselines.

**SOTA performance on length instruction following.** LARFT consistently achieves superior performance across all length-related metrics. Specifically, on LIFEBench, LARFT obtains the lowest LD and the highest LS across all four base models, demonstrating a significant enhancement in length instruction following across an extensive length constraints. Compared to the second-best method, the LS

improves by up to **12.30** points (on Qwen2.5-3B-Instruct). Additionally, on LongBench, which specifically evaluates long-generation capabilities, the length score ($S_l$) achieves the best performance, while the generation quality ($S_q$) shows no significant variation. Compared to the pure RL baseline, which also relies on self-exploration without external expert supervision, LARFT improves $S_l$ by **1.11**, **2.91**, **0.46**, and **4.36** across the four models, respectively. In terms of short-length instruction following, LARFT achieves the lowest MAE on three out of four models, only slightly lagging behind the pure RL baseline on Llama-3.1-8B-Instruct by a marginal gap of **0.41**. Meanwhile, it guarantees high response quality, maintaining ROUGE-L scores at either the first or second rank across all models.

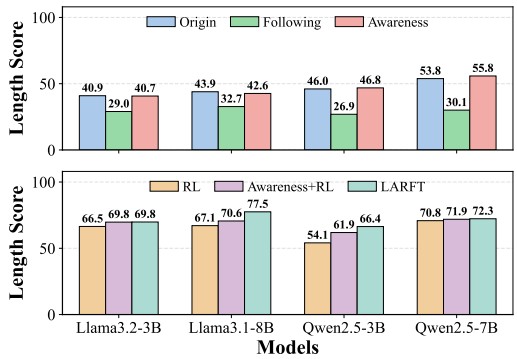

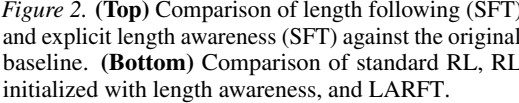

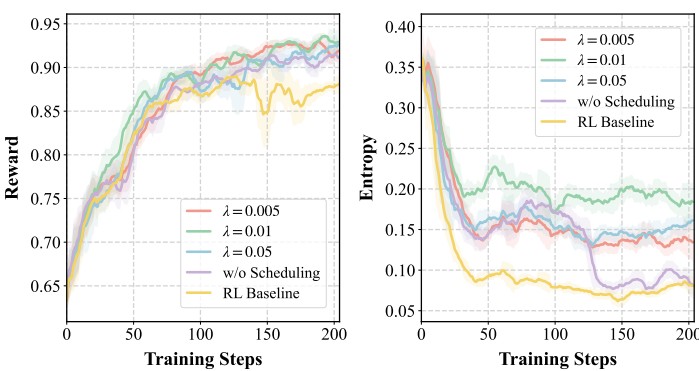

*Figure 2.* (**Top**) Comparison of length following (SFT) and explicit length awareness (SFT) against the original baseline. (**Bottom**) Comparison of standard RL, RL initialized with length awareness, and LARFT.

*Figure 3.* Ablation study of the unified coefficient $\lambda$. (**Left**) The plot shows the training reward over steps. (**Right**) The plot shows the entropy changes. Default setting $\lambda = 0.01$ demonstrates superior performance, while "w/o Scheduling" indicates the removal of the Cosine Annealing schedule.

**Robustness on general abilities.** We further examine out-of-distribution performance to evaluate whether length-targeted training compromises the model's general abilities. As shown in the right block of Table 1, LARFT demonstrates a minimal impact on general abilities across multiple benchmarks. Specifically, on datasets such as MMLU, GSM8K, and GPQA, the performance degradation is negligible, with the maximum drop observed being only **2.02** points (on GPQA with Qwen2.5-3B, decreasing from **31.03** to **29.01**). This confirms that our method effectively maintains the core cognitive abilities of the base models. Furthermore, regarding general instruction following as measured by IFEval, LARFT achieves the first or second rank on Qwen2.5-7B and both Llama models. In contrast, the second-best method, pure RL, suffers from varying degrees of performance regression across all tested models(e.g., dropping from **85.85** to **73.38** on Llama3.1-8B), highlighting the superiority of LARFT in balancing specific length constraints with general instruction following.

**The Inefficacy of Supervised Paradigms.** A critical observation is the significant limitation of traditional SFT-based paradigms when following length constraints across extended contexts. SFT based methods including standard SFT, PositionID and Ruler exhibit a drastic regression in length-following capabilities. While these models generally maintain or improve performance on long generation abilities (LongBench), their ability to follow short length constraints (Lenctrl-Bench) deteriorates sharply. On Lenctrl-Bench, all SFT-based paradigms witness an increase in MAE of over **15** points across all tested models, revealing a severe deficiency in following short-length instructions. We hypothesize that this failure stems from two primary factors. First, the dominant loss contribution from long-sequence generation in mixed training batches may dilute the model's attention to specific length constraints. Second, supervised learning relies solely on positive examples, lack-

ing the negative contrastive signals necessary for the model to distinguish precise boundaries of length compliance.

The results show that while SFT provides necessary expert priors regarding length, relying on it as a direct optimization target leads to the observed performance regression. LARFT effectively resolves this by utilizing SFT solely to inject external expert knowledge and employing RL as the definitive optimization objective. This synergy allows LARFT to leverage the advantages of both approaches, thereby outperforming existing baselines in length-following tasks.

### 5.2. The Role of Length Awareness

To investigate whether the explicit training of length awareness genuinely contributes to length instruction following and to identify the specific stage at which this enhancement occurs, we conducted a controlled experiment. Specifically, we applied the relabeling strategy (§3.2) to our datasets to train a model exclusively on length awareness tasks, establishing an *Awareness* baseline. Subsequently, we applied the length-oriented reinforcement learning (§3.1) to this awareness-initialized model. The comparative results are presented in Figure 2.

The results reveal a distinct "Cognition-Action Gap". In the SFT stage, the *Awareness* model exhibits mixed but minor fluctuations compared to the *Origin* baseline. Specifically, we observe a slight decline on Llama models (e.g., **40.9** → **40.7** on Llama3.2-3B), contrasted with a marginal increase on Qwen models (e.g., **46.0** → **46.8** on Qwen2.5-3B). These variations indicate that merely enhancing the internal representation of length (cognition) has no significant impact on the actual ability to follow length constraints(action).

A significant contrast emerges when RL is applied. The model initialized with length awareness consistently outperforms the standard RL baseline, achieving an average improvement of **3.93** points across four distinct backbones.

This substantial gain confirms that the model's action capability benefits from cognitive guidance. Nevertheless, it still lags behind LARFT with an average deficit of **2.95** points. This gap implies that cognitive degradation bottlenecks action improvement. LARFT overcomes this by synchronizing cognitive maintenance with action refinement, ensuring sustained capability gains.

### 5.3. Ablation on Auxiliary Task Design

To verify whether the performance gains stem specifically from the proposed *on-policy hindsight counting* mechanism rather than merely introducing extra compute or generic length-related regularization, we conduct a comprehensive ablation on Qwen2.5-7B across three benchmarks. We compare LARFT against five configurations: (1) RL only; (2) Compute-matched RL, introducing extra dummy forward passes to match LARFT's computational cost; (3) Off-policy counting SFT, applying the same counting loss but on a fixed external corpus rather than the model's own rollouts; (4) Long/short classification, a coarser auxiliary task predicting only binary length categories; and (5) Random auxiliary, a generic regularization loss without length semantics.

As shown in Table 2, introducing compute-matched extra steps yields only marginal improvements over the pure RL baseline. Surprisingly, applying counting supervision on off-policy data actually hurts performance (**LS** $= 66.31$**, worse than RL-only**), highlighting the cognitive gap between external text and the model's internal generation dynamics. Coarser signals like binary classification or random auxiliary losses also fail to provide meaningful gains. Only LARFT's on-policy hindsight counting produces substantial improvements, confirming that the specific design of converting the model's own rollouts into precise counting supervision is the critical driver of success.

### 5.4. Impact of Unified Optimization Mechanism

To validate the effectiveness of the unified optimization mechanism, we conduct an ablation study comparing different values of $\lambda_t$ and evaluating the performance without the cosine annealing strategy on Qwen2.5-7B.

**Importance of Unified Optimization.** Figure 3 illustrates a critical divergence: while the RL baseline and LARFT share similar early growth (0-50 steps), the baseline later plateaus with high variance, struggling to refine length-following capabilities solely via scalar rewards. In contrast, our unified optimization sustains a steady upward trajectory. Crucially, our method inherently promotes effective exploration: while the RL baseline suffers from rapid entropy collapse and premature convergence, LARFT maintains higher entropy levels. This behavior is akin to strategies in reasoning-oriented RL (He et al., 2025; Wang et al., 2025;

*Table 2.* Comparison between LARFT and four alternative auxiliary task designs on Qwen2.5-7B. The results demonstrate that our proposed on-policy hindsight design significantly outperforms the pure RL baseline and all alternative auxiliary configurations.

| Configuration | LS ↑ | $S_l$ ↑ | MAE ↓ |
|---|---|---|---|
| RL | 70.94 | 93.84 | 9.01 |
| + Compute-matched steps | 72.20 | 94.48 | 8.99 |
| + Off-policy counting SFT | 66.31 | 83.41 | 9.72 |
| + Long/short classification | 64.85 | 80.18 | 9.33 |
| + Random auxiliary | 60.92 | 78.12 | 15.95 |
| **LARFT** (On-policy hindsight) | **80.57** | **96.75** | **7.00** |

*Table 3.* Ablation study on optimization hyperparameters. We report the performance across three length-following benchmarks. "Sc" indicates Cosine Annealing Scheduling.

| Method | $\lambda_t$ | | | w/o Sc |
|---|---|---|---|---|
| | 0.005 | **0.01 (Default)** | 0.05 | |
| LIFEBench LS (↑) | 75.67 | **80.57** | 79.29 | 73.44 |
| LongBench $S_l$ (↑) | 94.58 | **96.75** | 96.12 | 94.22 |
| Lenctrl-Bench MAE (↓) | 8.45 | **7.00** | 7.36 | 8.80 |

Cui et al., 2025), where sustained exploration prevents local optima and drives superior performance. Notably, this advantage remains robust across different $\lambda$ values.

**Ablation of $\lambda$ and Cosine Annealing Scheduling.** Table 3 validates our hyperparameter choices and the necessity of the Cosine Annealing scheduling. While $\lambda = 0.01$ serves as the optimal default, the method demonstrates robustness to magnitude variations. Shifting $\lambda$ to 0.05 or 0.005 results in relatively stable performance (e.g., LS of 79.29 and 75.67, respectively), indicating that exact tuning is not overly sensitive. In contrast, removing the Cosine Annealing schedule ("w/o Cosine") leads to the most significant performance degradation, with the score plummeting from 80.57 to 73.44 on LIFEBench. This sharp decline highlights that the dynamic scheduling strategy is far more critical for success than the precise magnitude of the weight itself.

### 5.5. Mechanistic Interpretability Analysis

To determine whether the trained models have genuinely internalized length constraints, we conduct a length probing experiment designed to decode length representations directly from the model's hidden states. Specifically, we regress the actual length of the final response using hidden states from each layer at two critical timestamps: the *first generated token* and the *last generated token*. This approach allows us to probe the model's plan of generation prior to generation and its awareness post-generation. Figure 4 illustrates the correlation between fitted and actual lengths for Qwen2.5-7B, with detailed settings in Appendix D.

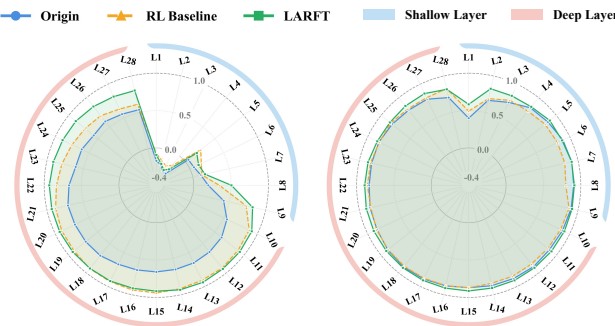

Origin  RL Baseline  LARFT  Shallow Layer  Deep Layer

*Figure 4.* Length probing correlations across layers. The plots compare the ability to decode response length from the hidden states of the first token (**Left**) and the last token (**Right**).

**Planning Capability.**   As shown in the left panel, probing results on the first token reveal that LARFT achieves the highest correlation, significantly outperforming baselines. A striking phenomenon observed across all models is the layer-wise distribution of this capability: probing accuracy is negligible in shallow layers but exhibits a sharp emergence around Layer 10. This observation suggests that length planning is a high-level cognitive function; the model requires deep-layer processing to abstract and formulate a global length strategy before the generation process initiates.

**Awareness Capability.**   For the last generated token (right panel), LARFT maintains an overall lead. In contrast to the planning capability, we observe that this awareness emerges remarkably early, achieving high accuracy as early as Layer 2 and persisting throughout the network. This indicates that the ability to count the length of the current context is a fundamental, low-level feature encoded in shallow layers. LARFT's superiority here implies that it not only optimizes deep-layer planning but also improves basic shallow-layer awareness, effectively internalizing the concept of length.

**Causal Role of Length Representations.**   While the probing results confirm that length information is encoded in hidden states, we further investigate whether the model actively relies on these representations to govern generation length. To establish a causal link, we conduct a directional ablation experiment on Qwen2.5-7B. Specifically, using the trained linear probe, we identify the top-50 dimensions most predictive of length across Layers 2–10, and zero out their activations during autoregressive generation. As shown in Table 4, while ablating randomly selected dimensions as a control yields a negligible effect, zeroing out the length-predictive dimensions causes a significant, graded degradation in Length Score (LS). Notably, LARFT experiences the most pronounced performance drop among all configurations. This confirms that LARFT not only learns a stronger representation of length, but builds internal structures that the model causally depends on to guide generation.

*Table 4.* Causal intervention via directional ablation on Qwen2.5-7B using LIFEBench.

| Method | Length-Predictive Dims | | Random Dims | |
|---|---|---|---|---|
| | LS ($\uparrow$) | $\Delta$ LS ($\downarrow$) | LS ($\uparrow$) | $\Delta$ LS ($\downarrow$) |
| Base | 51.18 | -2.66 | 52.87 | -0.97 |
| RL | 61.73 | -9.21 | 69.98 | -0.96 |
| LARFT | 63.08 | **-17.49** | 79.41 | -1.16 |

# 6. Conclusion

In this work, we improve length instruction following capabilities in LLMs with **LARFT**, a framework designed to bridge cognition and action. By integrating Length-Oriented Reinforcement Learning with Hindsight Length Awareness, LARFT enables models to effectively internalize and control output length. Empirical results demonstrate that our method achieves state-of-the-art performance without compromising general capabilities. Future work will explore the generalization of this framework to more diverse generation scenarios such as logical reasoning and stylistic generation.

## Acknowledgments

This work was supported by the National Key Research and Development Program of China (Grant No. 2024YFF0907401).

## Impact Statement

This paper presents work whose goal is to advance the field of machine learning. There are many potential societal consequences of our work, none of which we feel must be specifically highlighted here.

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

# A. Training Algorithm

This appendix provides the complete training procedure for LARFT, which integrates Length-Oriented Reinforcement Learning with Hindsight Length Awareness in a unified optimization framework.

## A.1. Algorithm Overview

Algorithm 1 presents the detailed pseudocode for the LARFT training procedure. The algorithm operates in an iterative fashion, where each training step consists of three main phases: (1) rollout generation and reward computation, (2) hindsight awareness data construction, and (3) unified loss optimization with dynamic weight scheduling.

---

**Algorithm 1** LARFT: Unified Training Procedure

---

**Require:** Pre-trained language model $\pi_\theta$, training dataset $\mathcal{X}$ with length constraints, total training steps $T$, group size $G$, maximum awareness weight $\lambda_{\max}$, clipping hyperparameter $\epsilon$
**Ensure:** Trained model $\pi_\theta$ with enhanced length instruction following capabilities
 1: Initialize policy parameters $\theta$ from pre-trained model
 2: Initialize replay buffer $\mathcal{D} \leftarrow \emptyset$
 3: **for** $t = 1$ **to** $T$ **do**
 4:     *// Phase 1: Rollout Generation & Reward Computation*
 5:     Sample batch of prompts $\{(x_j, c_j)\}_{j=1}^{B}$ from $\mathcal{X}$
 6:     **for** each prompt $(x_j, c_j)$ in batch **do**
 7:         Sample $G$ trajectories $\{\tau_{j,1}, \ldots, \tau_{j,G}\}$ from $\pi_{\theta_{\text{old}}}(\cdot|x_j)$
 8:         **for** $i = 1$ **to** $G$ **do**
 9:             Compute word count: $\ell_{j,i} \leftarrow L(\tau_{j,i})$
10:             Compute deviation: $\delta_{j,i} \leftarrow |\ell_{j,i} - c_j|/c_j$
11:             Compute reward: $R_{j,i} \leftarrow \max(0, 1 - \delta_{j,i})$
12:         **end for**
13:         Compute group statistics: $\mu_j \leftarrow \text{mean}(\{R_{j,i}\})$, $\sigma_j \leftarrow \text{std}(\{R_{j,i}\})$
14:         Compute advantages: $A_{j,i} \leftarrow (R_{j,i} - \mu_j)/(\sigma_j + \epsilon_{\text{std}})$
15:     **end for**
16:     *// Phase 2: Hindsight Awareness Data Construction*
17:     $\mathcal{D}_t \leftarrow \emptyset$                                                                                 *// Current step awareness batch*
18:     **for** each trajectory $\tau_{j,i}$ in current batch **do**
19:         Construct awareness input: $x_{\text{aware}} \leftarrow [\tau_{j,i}; \mathcal{I}_{\text{aware}}]$
20:         Construct target label: $y_{\text{aware}} \leftarrow \texttt{FormatLength}(\ell_{j,i})$
21:         $\mathcal{D}_t \leftarrow \mathcal{D}_t \cup \{(x_{\text{aware}}, y_{\text{aware}})\}$
22:     **end for**
23:     *// Phase 3: Unified Loss Computation*
24:     Compute GRPO loss $\mathcal{L}_{\text{GRPO}}$ using Eq. 5
25:     Compute awareness loss $\mathcal{L}_{\text{aware}}$ using Eq. 7 on $\mathcal{D}_t$
26:     *// Dynamic weight scheduling (Cosine Annealing)*
27:     $\lambda_t \leftarrow \frac{\lambda_{\max}}{2}\left(1 + \cos\left(\frac{t \cdot \pi}{T}\right)\right)$
28:     *// Unified optimization*
29:     $\mathcal{L}_{\text{unified}} \leftarrow \mathcal{L}_{\text{GRPO}} + \lambda_t \cdot \mathcal{L}_{\text{aware}}$
30:     Update $\theta$ by minimizing $\mathcal{L}_{\text{unified}}$ via gradient descent
31:     $\theta_{\text{old}} \leftarrow \theta$                                                                                  *// Update reference policy*
32: **end for**
33: **return** $\pi_\theta$

---

## A.2. Implementation Details

**Word Counting Function** The function $\text{COUNTWORDS}(\cdot)$, is implemented to handle both Chinese and English text uniformly. Let $s = \mathcal{D}(\tau)$ denote the detokenized string derived from the candidate sequence. For Chinese text, each character is counted as one word; for English text, standard whitespace-delimited tokenization is applied. Formally, we

define:

$$\text{COUNTWORDS}(s) = |\text{MATCH}(s, \mathcal{P})|, \tag{9}$$

where $\mathcal{P} = $ `[\u4e00-\u9fff]|[a-zA-Z0-9\'-]+` is a regex pattern that matches Chinese characters and English words.

**Awareness Instruction Format** The awareness instruction $\mathcal{I}_{\text{aware}}$ is designed as a structured prompt that directs the model to perform self-evaluation:

---

**Awareness Instruction Template**

```
You are a precise text analysis assistant.  Your task is to count the number of
words in the provided text, where words include both Chinese characters and English
words/numbers.
Respond ONLY with the final word count, enclosed in XML tags like this:
<word_count>XXX</word_count>.
-- TEXT TO ANALYZE --
{generated_response}
```

---

**Label Format** The ground-truth label for the awareness task follows a structured XML format:

$$y_{\text{aware}} = \text{<word\_count>}\|L(\tau)\|\text{</word\_count>} \tag{10}$$

where $\|$ denotes string concatenation. This structured format provides clear supervision signals and facilitates reliable parsing during evaluation.

### A.3. Training Dynamics

**Loss Masking Strategy** When computing the awareness loss $\mathcal{L}_{\text{aware}}$, we apply a masking strategy to ensure that gradients only flow through the label portion. Specifically, for the composite input $[x_{\text{aware}}; y_{\text{aware}}]$, the loss is computed only on tokens corresponding to $y_{\text{aware}}$:

$$\mathcal{L}_{\text{aware}} = -\sum_{k=|x_{\text{aware}}|+1}^{|x_{\text{aware}}|+|y_{\text{aware}}|} \log P_\theta(o_k|o_{<k}) \tag{11}$$

**Dynamic Weight Schedule Visualization** Figure 5 illustrates the evolution of the awareness weight $\lambda_t$ throughout training. The cosine annealing schedule ensures that in the **early stage** ($t \ll T$), $\lambda_t \approx \lambda_{\max}$ emphasizes length awareness to establish foundational representations, while in the **late stage** ($t \rightarrow T$), $\lambda_t \rightarrow 0$ prioritizes policy optimization for precise instruction following.

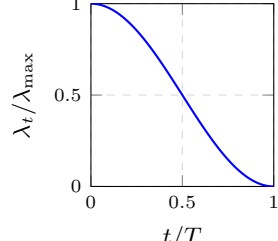

*Figure 5.* Cosine annealing schedule for $\lambda_t$.

### A.4. Computational Complexity

The unified training procedure introduces minimal computational overhead compared to standard GRPO. Since the rollout phase remains identical—requiring $G$ forward passes per prompt—and the awareness data construction involves negligible string operations ($O(B \cdot G)$), the primary additional cost arises from a single extra forward pass needed to compute $\mathcal{L}_{\text{aware}}$. Consequently, the overall complexity per training step scales as $O((G + 1) \cdot B \cdot |\tau|)$, where $|\tau|$ denotes the average trajectory length. This additional $(+1)$ term corresponds to a marginal overhead of approximately $\frac{1}{G+1}$ relative to pure GRPO training.

## B. Data Construction Pipeline

This appendix details the complete data construction pipeline for creating training samples where explicit length constraints are integrated with semantic instructions without conflicts.

## B.1. Data Sources

We source our raw data from two high-quality datasets distilled from DeepSeek-R1:

- **AM-DeepSeek-R1-Distilled-1.4M**: A large-scale English reasoning dataset containing 1.4 million samples with diverse task categories including instruction following, creative writing, and question answering.

- **Chinese-DeepSeek-R1-Distill-data-110k-SFT**: A Chinese SFT-format dataset containing approximately 110,000 samples covering open-domain QA, creative writing, and analysis tasks.

Together, these datasets provide over 1.5 million raw samples with bilingual coverage.

## B.2. Construction Pipeline

Our data construction pipeline comprises six sequential stages, as illustrated in Figure 6. Starting from 1.51M+ raw samples, we progressively filter and transform the data through rule-based filtering, content extraction, AI-assisted curation, instruction synthesis, and final quality filtering, ultimately yielding 8,732 high-quality training samples.

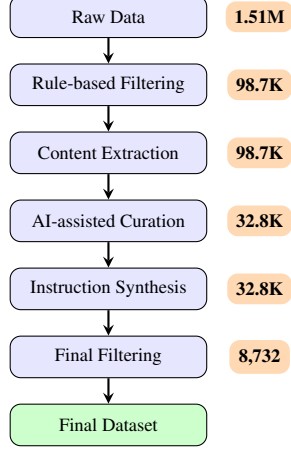

*Figure 6.* Data construction pipeline with sample counts at each stage.

### B.2.1. STAGE 1: RULE-BASED FILTERING

We first apply rule-based filters to exclude task types that are inherently incompatible with flexible length control. The filtering criteria include:

- **Repository-based exclusion**: We exclude samples from specific repositories containing mathematical problems (e.g., `Advanced-Math`, `GSM8K_zh`), exam questions (e.g., `coig_exam`, `kaoyan`), and domain-specific STEM content (e.g., `stem_zh/chem`, `stem_zh/phy`).

- **Quality score filtering**: For Chinese data, we retain only samples with quality scores $\geq 9$ (on a 10-point scale).

- **Content-type detection**: We implement regex-based detectors to exclude:
  - **Code snippets**: Detected via programming keywords (`def`, `class`, `import`), markdown code blocks, and indentation patterns.
  - **ASCII tables**: Identified by pipe-delimited row patterns.
  - **LaTeX formulas**: Detected via `$$...$$`, `\[...\]`, and common LaTeX commands.
  - **HTML content**: Identified by tag patterns and HTML entities.
  - **Non-target language**: For English data, we exclude samples containing Chinese, Japanese, Korean, or Russian characters.

This stage reduces the dataset from 1.51M to approximately 98.7K samples.

### B.2.2. STAGE 2: CONTENT EXTRACTION

The source datasets contain Chain-of-Thought (CoT) reasoning traces alongside final responses. We extract clean response content using the following procedure:

1. Remove thinking traces enclosed in `<think>...</think>` tags.

2. Extract content from `<response>` tags when present.

3. Strip assistant markers (e.g., `<|im_start|>assistant`, `<|im_end|>`).

### B.2.3. STAGE 3: AI-ASSISTED SAMPLE CURATION

We employ GPT-4.1 (OpenAI, 2025) to perform semantic filtering, identifying samples suitable for length-constrained rewriting. The AI evaluates each sample based on:

- **Task compatibility**: Whether the task type is suitable for exact length constraints (excluding classification, NER, fill-in-the-blank, multiple choice, etc.).

- **Content expandability**: Whether the response can naturally expand or compress without losing coherence.

- **Fixed-form exclusion**: Excluding tasks with inherently fixed output formats (poetry, couplets, idioms, slogans, etc.).

The detailed prompts used for AI-assisted curation are provided in Section B.3.

### B.2.4. STAGE 4: INSTRUCTION SYNTHESIS

The original prompts are rewritten by LLMs to explicitly integrate length constraints. The AI modifies each instruction to include an exact word/character count requirement based on the actual response length. This stage ensures that:

- Existing length constraints (if any) are converted to exact equality constraints.

- New length constraints are inserted naturally without changing the original meaning or tone.

- Only "equal to" constraints are used (no "at least", "at most", etc.).

### B.2.5. STAGE 5: FINAL QUALITY FILTERING

The final filtering stage applies additional quality checks on the AI-modified samples, including response coherence verification and length constraint consistency checks, yielding the final dataset of 8,732 samples.

### B.3. AI-assisted Curation Prompts

---

**Curation Prompt**

Please determine whether a user input prompt is suitable for imposing or modifying to a fixed-length output constraint—that is, requiring the generated output to match a specific length (not "at least", "at most", "no more than", "no less than", etc.).
Follow these guidelines:
1. If the original prompt already contains any length, character, or word count requirements, proactively modify all such requirements to use an "{length}{unit}" length constraint.
2. If appropriate, and without changing the original meaning, tone, or adding any new requirements except for the explicit length constraint, insert an "{length}{unit}" length requirement at a natural position in the prompt.
3. Only use exact equality for the length constraint—do not allow expressions like "within", "at least", "no more than", "no less than", etc.
4. If the original prompt falls into any of these categories, it is NOT suitable: classification, POS tagging, NER, fill-in-the-blank, multiple choice, true/false, labeling, annotation, matching, ranking, scoring, synonym/antonym selection, error correction, information extraction, etc.
5. If the original prompt is for a task with inherently fixed/short creative expression, it is NOT suitable: poetry, couplets, riddles, lyrics, famous quotes, slogans, advertisements, titles, idioms, proverbs, etc.
6. For open-ended generation, creative writing, analytical or reasoning tasks, it is generally acceptable to add an exact length constraint.
Output in JSON format: explain (reasoning), suitable (boolean), modified_prompt (revised prompt if suitable).

---

### B.4. Dataset Statistics

The constructed dataset consists of 8,732 samples, exhibiting a diverse distribution of sequence lengths ranging from 10 to 4,000 words. As detailed in Table 5, the data is categorized into four distinct groups based on word count. The distribution is relatively balanced across short, medium, and long sequences, ensuring the model is exposed to a wide variety of context lengths during training.

*Table 5.* Distribution of dataset samples by length category. The dataset covers a broad range of lengths, with the majority of samples falling into the Medium (101–500 words) category.

| Length Category | Word Range | Count | Percentage |
|---|---|---|---|
| Short | 10–100 | 2,156 | 24.7% |
| Medium | 101–500 | 3,421 | 39.2% |
| Long | 501–1,000 | 1,892 | 21.7% |
| Very Long | 1,001–4,000 | 1,263 | 14.5% |
| **Total** | **10–4,000** | **8,732** | **100.0%** |

## C. Evaluation Settings

This appendix provides comprehensive details on the evaluation benchmarks, metrics, and inference configurations used in our experiments.

### C.1. Length Instruction Following Benchmarks

We evaluate length instruction following capabilities using three complementary benchmarks that cover different length ranges and evaluation paradigms.

#### C.1.1. LIFEBENCH

LIFEBench (Zhang et al., 2025a) is a comprehensive length instruction following benchmark covering a wide range of length constraints. We use two primary metrics:

- **Length Score (LS)**: Measures the percentage of responses that fall within an acceptable deviation from the target length. Higher is better.

- **Length Deviation (LD)**: Computes the average relative deviation between actual and target lengths. Lower is better.

We restrict evaluation to samples with target lengths under 4,000 words due to the maximum output length limitations of the base models. This filtering removes approximately 10% of the original test set while ensuring fair comparison across all model sizes.

#### C.1.2. LONGBENCH

LongBench (Bai et al., 2024) evaluates long-form generation capabilities with length constraints ranging from 500 to 10,000+ words. We use the `longbench_write.jsonl` subset and report two metrics:

- **Length Score** ($S_l$): A normalized score measuring adherence to length requirements, computed as:

$$S_l = \begin{cases} 100 \cdot \max\left(0, 1 - \frac{y/x - 1}{3}\right) & \text{if } y > x \\ 100 \cdot \max\left(0, 1 - \frac{x/y - 1}{2}\right) & \text{if } y \leq x \end{cases} \tag{12}$$

  where $x$ is the required length and $y$ is the actual output length. This asymmetric scoring penalizes under-generation more heavily than over-generation.

- **Quality Score** ($S_q$): Using an LLM-as-Judge evaluation to evaluate response quality across six dimensions: *Relevance*, *Accuracy*, *Coherence*, *Clarity*, *Breadth and Depth*, and *Reading Experience*. Each dimension is scored on a 1–5 scale, and the final quality score is the normalized average.

**LLM-as-a-Judge Configuration.** To ensure robust and consistent quality evaluation, we employ **DeepSeek-V3** (Liu et al., 2024a) as our evaluator model. We configure the generation parameters with a temperature of 0.5 to strike a balance between diversity and determinism, and set the maximum new token limit to 1,024 to accommodate detailed critiques.

## C.1.3. LENCTRL-BENCH

Lenctrl-Bench (Wang et al., 2024) focuses on short-form length constraints (typically under 500 words). We report:

- **MAE (Mean Absolute Error)**: The average absolute difference between target and actual word counts. Lower is better.

- **ROUGE-L**: Measures response quality by computing the longest common subsequence overlap with reference responses. Higher is better.

## C.2. General Capability Benchmarks

To ensure that length-targeted training does not compromise general model capabilities, we evaluate on four widely-used benchmarks using the `lm-evaluation-harness` framework (Gao et al., 2023).

*Table 6.* General capability benchmark configurations.

| Benchmark | Task Type | Few-shot | Metric |
|---|---|---|---|
| MMLU (Hendrycks et al., 2021) | Knowledge | 5-shot | Accuracy |
| GSM8K (Cobbe et al., 2021) | Math Reasoning | 5-shot | Exact Match |
| IFEval (Zhou et al., 2023) | Instruction Following | 0-shot | Prompt Strict Acc |
| GPQA (Rein et al., 2024) | Graduate-level QA | 5-shot | Accuracy |

## C.3. Inference Configuration

All evaluations use consistent inference settings across models and benchmarks to ensure fair comparison.

*Table 7.* Inference configuration for all evaluations.

| Parameter | Length Benchmarks | General Benchmarks |
|---|---|---|
| Backend | vLLM | vLLM |
| Temperature | 0.6 | 0.0 (greedy) |
| Top-p | 0.9 | 1.0 |
| Max New Tokens | 8192 | Task-dependent |
| Tensor Parallel Size | 4 | 4 |

**Hardware**   All inference is conducted on a cluster with $8\times$ NVIDIA A100 80GB GPUs. We use 4-way tensor parallelism for efficient inference across all model sizes.

# D. Mechanistic Interpretability Analysis

This appendix details the methodology for the mechanistic interpretability analysis. We employ linear probing to investigate how length information—specifically the model's internal planning and real-time awareness—is encoded across different layers of the network.

## D.1. Linear Probing Methodology

Linear probing is a standard technique for analyzing the information content of neural representations (Alain & Bengio, 2017; Belinkov et al., 2017). We train linear regressors on frozen hidden states to determine if the *actual length* of the generated response is linearly decodable from the model's activation space.

**Theoretical Formulation**   Let $\mathbf{h}_l \in \mathbb{R}^d$ denote the hidden state vector extracted from layer $l$ at a specific token position. We aim to train a linear function $f(\mathbf{h}_l) = \mathbf{w}^\top \mathbf{h}_l + b$ to predict the scalar value $y$, which represents the actual length of the model's final response.

To quantify the strength of this encoding, we calculate the Pearson Correlation Coefficient ($r$) between the predicted lengths $\hat{y}$ and the actual lengths $y$:

$$r = \frac{\sum(y_i - \bar{y})(\hat{y}_i - \bar{\hat{y}})}{\sqrt{\sum(y_i - \bar{y})^2 \sum(\hat{y}_i - \bar{\hat{y}})^2}} \tag{13}$$

A high correlation ($r \to 1$) implies that the layer possesses a strong linear representation of the length information, effectively "knowing" how long the sequence will be (planning) or currently is (awareness).

### D.2. Experimental Setup

**Model Variants**    Consistent with the main experiments, we analyze the **Qwen2.5-7B-Instruct** architecture (28 transformer layers) across three settings:

- **Baseline**: The original supervised fine-tuned model.

- **RL Only**: The model trained via GRPO with length rewards but without hindsight tokens.

- **LARFT**: Our proposed method integrating RL with hindsight length awareness.

**Probing Dataset**    We construct a probing dataset by randomly sampling 100 prompts from LIFEBench, covering a diverse range of target lengths (128–512 words) in both Chinese and English. For each prompt, we generate a response using the respective model and record the *actual generated length* as the regression label $y$. This ensures we are probing the model's internal state relative to its own behavior, rather than its understanding of the external instruction.

**Probe Training**    We employ Ridge Regression with L2 regularization ($\alpha = 1.0$) to mitigate overfitting on the limited sample size. We report the average correlation scores obtained via 5-fold cross-validation.

### D.3. Probing Paradigms

To disentangle the model's capability to plan ahead from its ability to track current progress, we conduct probing at two distinct timestamps, as discussed in the main text:

#### D.3.1. PLANNING CAPABILITY (FIRST GENERATED TOKEN)

**Definition**    We extract hidden states at the position of the **first generated token** ($t = 1$). At this step, the model has processed the entire prompt but has not yet generated the content of the response.

**Interpretation**    Probing at this position tests the model's *ex-ante* planning capability. A high correlation with the final response length suggests that the model has formulated a global strategy and implicitly decided on the response length before generation commences. As observed in our results, this is a high-level cognitive function that primarily emerges in deeper layers (around Layer 10).

#### D.3.2. AWARENESS CAPABILITY (LAST GENERATED TOKEN)

**Definition**    We extract hidden states at the position of the **last generated token** ($t = T$), i.e., the End-of-Sequence (EOS) token or the final token before termination.

**Interpretation**    Probing at this position tests the model's *ex-post* awareness or counting capability. Since the generation is complete, the hidden state should ideally reflect the total length accumulated so far. Our analysis shows this is a fundamental feature encoded early in the network (starting from Layer 2), indicating that the model maintains a running count of the context length throughout the generation process.

## E. Hyperparameter Configurations

This appendix provides comprehensive hyperparameter configurations for all experiments.

## E.1. Model Specifications

We conduct experiments on four instruction-tuned models from two families, as shown in Table 8.

*Table 8.* Model specifications used in experiments.

| Model Family | Model | Params | Context | Maximum Output Length |
|---|---|---|---|---|
| Qwen2.5 | Qwen2.5-3B-Instruct | 3B | 32K | 8192 |
| | Qwen2.5-7B-Instruct | 7B | 32K | 8192 |
| Llama3 | Llama-3.2-3B-Instruct | 3B | 128K | 128K |
| | Llama-3.1-8B-Instruct | 8B | 128K | 128K |

## E.2. Training Hyperparameters

Tables 9 and 10 present the hyperparameters for SFT baseline and Length-Aware Reinforcement Fine-Tuning-specific configurations, respectively.

*Table 9.* SFT training hyperparameters.

| Hyperparameter | Value |
|---|---|
| Global Batch Size | 64 |
| Learning Rate | $1 \times 10^{-5}$ |
| LR Schedule | Cosine Decay |
| Warmup Ratio | 0.1 |
| Weight Decay | 0.01 |
| Max Seq Length | 8192 |
| Training Epochs | 3 |
| Optimizer | AdamW |
| Precision | bfloat16 |

*Table 10.* Length-Aware Reinforcement Fine-Tuning-specific hyperparameters.

| Hyperparameter | Value |
|---|---|
| *Awareness Scheduling* | |
| $\lambda_{\max}$ | 0.01 |
| Warmup | Linear (10%) |
| Decay | Cosine |
| *Base Configuration* | |
| RL Backend | GRPO |
| (See Table 11) | |

Table 11 details the GRPO reinforcement learning configuration used by both RL Only and Length-Aware Reinforcement Fine-Tuning.

*Table 11.* Hyperparameters for GRPO-based reinforcement learning.

| Hyperparameter | Value | Hyperparameter | Value |
|---|---|---|---|
| *Data Configuration* | | *Rollout Configuration* | |
| Rollout Batch Size | 128 | Rollouts per Prompt ($G$) | 4 |
| Max Prompt Length | 2048 | Temperature | 0.7 |
| Max Response Length | 8000 | Top-p | 0.8 |
| Truncation Side | Left | GPU Mem Utilization | 0.8 |
| *Policy Optimization* | | *Regularization & Schedule* | |
| Algorithm | GRPO | KL Coefficient ($\beta$) | 0.001 |
| Actor Learning Rate | $1 \times 10^{-6}$ | Entropy Coefficient | 0.01 |
| PPO Mini Batch Size | 32 | Total Epochs | 3 |

# F. Extended Experimental Results

In this section, we provide additional experimental results to further demonstrate the stability, generalization, and superiority of LARFT, addressing comprehensive baselines and sub-category performance.

## F.1. Multi-Seed Stability

To verify the training stability of our framework, we conduct repeated experiments using 3 random seeds on the Qwen2.5-7B backbone. For each trained checkpoint, we evaluate it 3 times across all three length-following benchmarks to account for variance in sampling-based generation. As shown in Table 12, the training variance is minimal across all metrics. LARFT's advantage over the RL baseline holds consistently across seeds, with non-overlapping confidence intervals on every benchmark, indicating highly stable improvements.

*Table 12.* Multi-seed evaluation on Qwen2.5-7B. We report mean $\pm$ standard deviation across 3 random seeds $\times$ 3 evaluations.

| Method | LIFEBench | | LongBench | | Lenctrl-Bench | |
|---|---|---|---|---|---|---|
| | LD ($\downarrow$) | LS ($\uparrow$) | $S_l$ ($\uparrow$) | $S_q$ ($\uparrow$) | MAE ($\downarrow$) | ROUGE-L ($\uparrow$) |
| RL | $17.46 \pm 0.94$ | $70.61 \pm 1.37$ | $93.58 \pm 0.97$ | $75.49 \pm 0.73$ | $9.01 \pm 0.49$ | $21.31 \pm 0.1$ |
| **LARFT** | $10.96 \pm 0.61$ | $80.18 \pm 0.58$ | $96.98 \pm 0.43$ | $82.77 \pm 0.66$ | $7.12 \pm 0.31$ | $21.06 \pm 0.12$ |

## F.2. Extended General Capability & Additional Baselines

We further evaluate LARFT on coding (HumanEval) and multi-turn dialogue (MT-Bench). We also include comparisons with two recent length-control methods, Hansel (Song et al., 2025) and MarkerGen (Yuan et al., 2025a) (reimplemented based on their papers due to no public code release).

As shown in Table 13 and 14, LARFT preserves specialized general capabilities with negligible degradation, while significantly outperforming both Hansel and MarkerGen across all length-following constraints.

*Table 13.* Extended general capabilities on Qwen2.5-7B.

| Method | HumanEval | MT-Bench |
|---|---|---|
| Base | 0.8476 | 7.92 |
| **LARFT** | 0.8390 | 7.88 |

*Table 14.* Comparison with additional baselines on Qwen2.5-7B.

| Method | LIFEBench | LongBench | Lenctrl |
|---|---|---|---|
| Hansel | 68.90 | 88.10 | 13.60 |
| MarkerGen | 65.80 | 84.60 | 15.90 |
| **LARFT** | **80.57** | **96.75** | **7.00** |

