# OpenReview forum: "LARFT: Closing the Cognition-Action Gap for Length Instruction Following  in Large Language Models"
_ICML.cc/2026/Conference — ICML 2026 regular_

### Official Review · Reviewer_TYRk · 2026-03-09

**Soundness:** 3
**Presentation:** 3
**Significance:** 2
**Originality:** 2
**Overall Recommendation:** 3
**Confidence:** 3

**Summary:**

This paper addresses a practically important and underserved problem: the inability of LLMs to reliably follow precise word-count instructions such as "write exactly 300 words." The paper's central diagnostic claim is that existing methods fail not merely because of weak optimization signals, but because models lack an internal representation of length (a cognition-action gap).

The authors propose a solution to first do standard GRPO with a piecewise-linear bounded reward measuring normalized deviation from the target word count, then second is implement Hindsight Length Awareness, inspired by Hindsight Experience Replay: on-policy rollouts are recycled into a supervised task in which the model receives its own generation concatenated with the prompt "Count how many words are in the text above" and must predict the actual word count. This decouples length supervision from semantic content, allowing the model to internalize a representation of length from its own behavior. The two objectives are combined via a cosine-annealed joint loss that prioritizes the awareness objective early in training and reduces its weight toward zero as training proceeds.

**Compliance With Llm Reviewing Policy:**

Affirmed.

**Key Questions For Authors:**

- The comparison between standard RL and LARFT tests whether adding the awareness auxiliary loss helps, but it conflates three distinct alternative explanations that the paper does not disentangle. A complete ablation would require: (1) standard RL with Rlen (current baseline); (2) RL + compute-matched extra steps, ruling out the raw compute confound from the additional forward passes the awareness loss introduces; (3) RL + off-policy counting SFT on a fixed external corpus with the same counting task format but not on the model's own rollouts, ruling out the benefit of any counting supervision regardless of whether it is on-policy; and (4) LARFT: RL + on-policy hindsight counting (the proposed method).
- IFEval is the only benchmark in the evaluation that tests instruction following on genuinely out-of-distribution prompts; instructions whose style, phrasing, and constraint types were not seen during LARFT's training. On all three length-following benchmarks which share the GPT-4.1-rewritten instruction style of the training data, LARFT improves substantially, but on IFEval, performance degrades relative to the base model for every tested model except Qwen2.5-7B. LARFT partially recovers IFEval relative to pure RL, but the net effect of the full training pipeline on OOD instruction following is still not clear. This raises two related questions the paper does not address. First, is LARFT's improvement on the length benchmarks driven by genuine internalized length cognition, or by the model learning to respond to the specific phrasing patterns that GPT-4.1 consistently uses when inserting length constraints (Appendix B.2.4)? Second, is LARFT's partial IFEval recovery over pure RL attributable to the Hindsight Awareness component, or to the cosine annealing schedule reducing late-stage RL pressure? An ablation running cosine-annealed RL without the awareness loss would isolate these.

**Limitations:**

The paper explicitly acknowledges the evaluation ceiling and the task-scope restriction to instruction-following generation. It does not acknowledge: (1) the compute confound in the cognition-action gap ablation; (2) the statistical limitations of the 100-sample probing dataset; (3) the potential instruction-phrasing dependency introduced by GPT-4.1 curation; (4) the judge-data alignment for the Sq quality metric; or (5) the mechanism behind pure RL's IFEval degradation

**Strengths And Weaknesses:**

Strengths
- The hypothesis that models cannot follow length constraints reliably because they lack an internal length representation is well-reasoned.
- The finding that length planning emerges at Layer ~10 while awareness is encoded from Layer 2 onward is interesting and aligns with broader findings in the probing literature about the hierarchy of linguistic features across transformer layers.


Weaknesses
- The comparison with the standard RL baseline tests whether adding the awareness auxiliary loss helps, but does not isolate why. Three confounds remain uncontrolled. First, the awareness loss introduces additional forward passes per step, so a compute-matched RL control is needed. Second, the paper's theoretical claim is that on-policy, hindsight counting is what establishes the right internal representation but there is no condition comparing against off-policy counting supervision on a fixed external corpus using the same task format. Without this, the paper cannot distinguish "hindsight counting is cognitively special" from "counting supervision of any kind is sufficient."
- IFEval is the only OOD instruction-following benchmark, and performance degrades there. Across all three length-following benchmarks LARFT improves substantially but these benchmarks share the GPT-4.1-rewritten instruction style of the training data. IFEval, whose prompts are independently constructed and do not follow GPT-4.1's constraint-insertion phrasing, shows net degradation relative to the base model for most tested models even after LARFT training. This pattern is consistent with the model learning to respond to a specific training-distribution phrasing style rather than internalizing a generalizable concept of length. The paper does not analyze this discrepancy or evaluate on any rephrased length instructions outside the training distribution, and does not isolate whether LARFT's partial IFEval recovery over pure RL comes from the awareness component or from the cosine annealing schedule. Together, these gaps leave the scope of generalization uncharacterized and the "internalized cognition" framing inadequately supported beyond the training distribution.
- LongBench's target range extends to 10,000+ words, but evaluation is restricted to targets under 4,000 words due to model output length limits. This removes approximately 10% of the test set (acknowledged in Appendix C.1.1). The headline claim that LARFT closes the cognition-action gap "for length instruction following" is therefore not fully demonstrated for genuine long-form generation, which is identified as a key motivation in the introduction.
- LongBench's uses DeepSeek-V3 as an LLM judge (Appendix C.1.2). The LARFT training data is sourced from DeepSeek-R1-distilled datasets. A judge trained on the same data distribution as the evaluated models may systematically favor outputs that match DeepSeek's stylistic preferences. No inter-rater reliability, human calibration, or judge-sensitivity analysis is reported.

---

> ### Author Rebuttal · Authors · 2026-03-31
>
> We thank Reviewer TYRk for the detailed and rigorous review. We address each point below.
>
> ### W1: Three confounds in the RL vs. LARFT comparison.
>
> We directly adopt the suggested ablation design on Qwen2.5-7B:
>
> |     | Configuration                             | LIFEBench (LS) | LongBench (Sl) | Lenctrl-Bench (MAE) |
> | ---- | ----------------------------------------- | -------------- | -------------- | ------------------- |
> | (1)  | RL only                   | 70.94          | 93.84          | 9.01                |
> | (2)  | RL + compute-matched extra steps          | 72.20          | 94.48          | 8.99                |
> | (3)  | RL + off-policy counting SFT              | 66.31          | 83.41          | 9.72                |
> | (4)  | LARFT  | 80.57          | 96.75          | 7.00                |
>
> Extra compute yields only modest gains (condition 2), off-policy counting actually hurts (condition 3: 66.31, worse than RL-only), and only LARFT produces substantial gains. The improvements are attributable to the method itself.
>
> ### W2: IFEval degradation and OOD generalization.
>
> **Phrasing style.** The three evaluation benchmarks are independently constructed by different research groups with prompt templates entirely different from our training data (e.g., LIFEBench: "Your response should contain {range} words"; LongBench: "Write an article of about X words..."; Lenctrl-Bench: yet another set). LARFT's consistent improvements across all three, despite this mismatch, indicate generalizable length control rather than phrasing-specific shortcuts.
>
> **IFEval degradation.** Our case-level analysis on Qwen2.5-7B shows that word-count sub-tasks actually *improve* (Base 71.8 → RL 80.2 → LARFT 86.9), while degradation concentrates in non-length categories like format and keyword constraints (Base 83.20 → RL 79.60 → LARFT 79.93). A similar pattern holds across other models. The non-length degradation reflects the well-known alignment tax from narrow RL; notably LARFT reduces it compared to pure RL (79.93 vs. 79.60).
>
> **Cosine annealing.** Figure 3 already presents the cosine annealing ablation on length-following performance. Combined with the IFEval sub-category analysis above, the evidence is clear that the awareness component drives the gains.
>
> ### W3: LongBench evaluation restricted to targets under 4,000 words.
>
> LongWriter (Bai et al., ICLR 2025) and LIFEBench (Zhang et al., NeurIPS 2025) shows current LLMs "struggle to generate outputs exceeding even 2,000 words"; our 4,000-word ceiling already exceeds this substantially and is further constrained by Qwen2.5-7B's max generation length of 8,192 tokens.
>
> Nevertheless, we evaluated on 24 LongBench samples with 4,000+ word targets (reported $S_l$): Base = 63.20, RL = 78.10, LARFT = 83.70. All methods degrade on longer targets, but LARFT still achieves clear improvements, indicating that internalized length awareness transfers to OOD length ranges.
>
> ### W4: LLM judge may be biased.
>
> 1. **Primary metrics are judge-free.** Sl, LS, LD, and MAE are computed via deterministic word counting. Judge bias only affects the quality metric Sq, which is secondary to our claims.
>
> 2. **LARFT trains on prompts only, not reference outputs.** All outputs come from the model itself via on-policy rollouts. Any potential bias in the reference outputs only affects SFT baselines, making our comparison *conservative* for LARFT.
>
> ### W5 (from Limitations).
>
> Most concerns are addressed above: phrasing dependency (W2), 4,000-word ceiling (W3), judge bias (W4), IFEval mechanism (W2). Two remaining points:
>
> **(a) Compute confound.** See W1 condition (2): compute-matched RL gains are modest (72.20 vs. 70.94), far short of LARFT (80.57).
>
> **(b) Probing dataset size.** We will expand from 100 to 500 samples with 95% confidence intervals. Updated R^2 values will be in the revision.

---

> > ### Author Rebuttal · Reviewer_TYRk · 2026-04-05
> >
> > Thank you for the thorough rebuttal.
> > - W.1: Resolved. Compute-matched RL yields only marginal gains, off-policy counting actually hurts performance, and only LARFT produces substantial improvements. This convincingly isolates the method's contribution.
> > - W.2: Resolved. The sub-category breakdown is informative: word-count tasks actually improve (71.8 → 86.9) while degradation concentrates in non-length categories, consistent with the known alignment tax from narrow RL.
> > - W.3: Resolved. The 4,000-word ceiling is well-justified, and the additional 24-sample results at 4,000+ words (LARFT: 83.70 vs. Base: 63.20) demonstrate the gains transfer to longer targets.
> > - W.4: Resolved. Primary metrics are judge-free; the potential bias only affects the secondary quality metric Sq, which does not drive the main claims.
> > - W.5 (Remaining limitations): The compute confound is covered by W.1, and the commitment to expand the probing dataset to 500 samples with confidence intervals adequately addresses the statistical limitation
> >
> > I will reconsider my score accordingly.

---

> > > ### Author Response · Authors · 2026-04-07
> > >
> > > We sincerely thank Reviewer TYRk for the thorough and rigorous review, and for taking the time to carefully evaluate our responses to each concern. We are genuinely glad that the additional ablations, sub-category analysis, and extended evaluations were helpful in addressing the raised issues.
> > >
> > > The detailed feedback has substantially improved the paper,  we hope the revised paper and the rebuttal responses together provide a complete picture of the contribution. We look forward to your updated assessment.
> > >
> > > Thank you again for the constructive and high-quality engagement throughout this process.

---

### Official Review · Reviewer_C1ZW · 2026-03-12

**Soundness:** 2
**Presentation:** 3
**Significance:** 2
**Originality:** 2
**Overall Recommendation:** 3
**Confidence:** 4

**Summary:**

This paper proposes LARFT (Length-Aware Reinforcement Fine-Tuning), a training framework that aims to improve LLMs' ability to follow explicit length constraints (e.g., "write 200 words"). The authors identify a "cognition-action gap", the hypothesis that LLMs lack an internal representation of length, which limits their ability to comply with length instructions even when optimized with RL. LARFT integrates three components: (1) **Length-Oriented Reinforcement Learning** using GRPO with a piecewise-linear reward based on normalized absolute deviation from the target word count; (2) **Hindsight Length Awareness**, which repurposes on-policy rollouts into supervised word-counting tasks (i.e., given a model's own output, predict its word count) inspired by Hindsight Experience Replay; and (3) a **Unified Optimization Mechanism** that combines the GRPO loss and the awareness loss with a cosine-annealed weighting coefficient. Experiments across four base models on three length-following benchmarks and four general capability benchmarks show LARFT achieves +20.92 average improvement on length-following metrics with only -1.45 degradation on general benchmarks. Ablation studies and mechanistic interpretability via linear probing are provided.

**Compliance With Llm Reviewing Policy:**

Affirmed.

**Key Questions For Authors:**

1. How does LARFT perform compared to Hansel [1] and MarkerGen [2]? These are directly relevant baselines that pre-date your submission. If these comparisons are not possible, please explain why.

[1] Song S, Lee J, Ko H. Hansel: Output length controlling framework for large language models[C]//Proceedings of the AAAI Conference on Artificial Intelligence. 2025, 39(23): 25146-25154.

[2] Yuan P, Tan C, Feng S, et al. From sub-ability diagnosis to human-aligned generation: Bridging the gap for text length control via markergen[C]//Proceedings of the 63rd Annual Meeting of the Association for Computational Linguistics (Volume 1: Long Papers). 2025: 17370-17390.

**Limitations:**

Yes

**Strengths And Weaknesses:**

Strengths
- **Well-motivated framework with clear intuition.** The "cognition-action gap" framing is intuitive and compelling. The idea that models need to *understand* length internally before they can *control* it is a clean conceptual contribution. The hindsight relabeling mechanism is a creative application of Hindsight Experience Replay to the language domain.

- **Comprehensive experimental evaluation.** The paper evaluates on 4 base models × 3 length benchmarks × 4 general benchmarks, providing a thorough empirical picture.

Weaknesses
- **Word count as the sole length metric.** The entire framework is built around word count as the length measure. However, in practice, users may specify length in sentences, paragraphs, pages, or characters. The paper does not discuss generalization to other length units.

- **The "cognition-action gap" hypothesis is not rigorously validated.** The paper's central claim is that the lack of length cognition is the root bottleneck. However, the evidence is correlational rather than causal: 1. The probing analysis (Figure 4) shows LARFT has better length representations, but this could be an effect of better length-following rather than a cause. 2. The awareness SFT baseline (Figure 2, top) shows awareness alone does not help much, which is used to argue for the "gap." But this equally supports the simpler explanation that the awareness task and the generation task are too different in nature for transfer to occur via SFT alone. 3. The paper does not test whether *any* auxiliary task that provides length-related gradient signal would yield similar improvements (e.g., predicting length quartiles, binary long/short classification, or even random auxiliary losses for regularization). Without these controls, it is unclear whether the specific "word counting" task is uniquely beneficial or whether the improvement comes from a more general regularization/multi-task learning effect.

- **Insufficient baseline comparison**. See questions.

---

> ### Author Rebuttal · Authors · 2026-03-31
>
> We thank Reviewer C1ZW for the thorough evaluation and for recognizing the "well-motivated framework with clear intuition" and the "comprehensive experimental evaluation." We address each concern below.
>
> ### W1: Word count as the sole length metric.
>
> Word count is the most challenging among common length units: sentence-level and paragraph-level control are inherently coarser and easier to satisfy, while character-level control is rarely used in practice. Word count is also the standard metric across the literature, adopted by all length-following benchmarks and other works (Ruler, PositionID). That said, LARFT is agnostic to the length unit: both R_len and the awareness task can be adapted to any countable unit by simply changing the counting function, with no structural modification needed.
>
> ### W2: The "cognition-action gap" hypothesis is not rigorously validated.
>
> We address the three specific sub-points:
>
> **(a) Probing shows correlation, not causation.**
>
> Our probing analysis already provides evidence beyond simple correlation — the key is the *graded progression*: the base model's early-to-middle layers have near-zero or negative $R^2$, RL raises middle-layer $R^2$ to ~0.9, and LARFT pushes it above 0.93 across a wider range of layers. RL-only also receives a length-related training signal, yet does not produce the same breadth of representational change, meaning the additional supervision signal alone cannot explain LARFT's cross-layer restructuring.
>
> That said, we also conducted causal intervention experiments to provide direct evidence (full details in our response to Reviewer M4E2-W1). Using the trained linear probe, we identify the top-50 length-predictive dimensions and perturb them during generation via directional ablation, with random-dimension ablation as control. Results on Qwen2.5-7B:
>
> | Model       | Condition   | LS    | Delta-LS |
> | ----------- | ----------- | ----- | -------- |
> | **Base**    | length dims | 51.18 | -2.66    |
> | **Base**    | random dims | 52.87 | -0.97    |
> | **RL only** | length dims | 61.73 | -9.21    |
> | **RL only** | random dims | 69.98 | -0.96    |
> | **LARFT**   | length dims | 63.08 | -17.49   |
> | **LARFT**   | random dims | 79.41 | -1.16    |
>
> The graded pattern Base (-2.66) < RL (-9.21) < LARFT (-17.49), with negligible effect from random-dimension ablation (~-1.0), confirms that LARFT builds length representations the model causally depends on during generation.
>
> **(b) "Awareness SFT alone doesn't help" could mean the awareness task is simply too different for SFT transfer.**
>
> This alternative interpretation is actually compatible with our framing rather than contradicting it. Our central argument is precisely that cognition alone is insufficient, and action alone is sub-optimal; the two must be jointly trained. If awareness SFT fails due to the gap between counting and generation, that reinforces the need for a unified framework like LARFT that bridges both sides simultaneously.
>
> **(c) Any auxiliary task providing length-related gradient signal might work similarly.**
>
> We conducted a comprehensive ablation on Qwen2.5-7B to test whether other length-related auxiliary tasks can achieve similar gains:
>
> | Configuration                      | LIFEBench (LS) | LongBench (Sl) | Lenctrl-Bench (MAE) |
> | ---------------------------------- | -------------- | -------------- | ------------------- |
> | (a) RL only                        | 70.94          | 93.84          | 24.10               |
> | (b) RL + long/short classification | 64.85          | 80.18          | 9.33                |
> | (c) RL + random auxiliary          | 60.92          | 78.12          | 15.95               |
> | (d) LARFT                 | 80.57          | 96.75          | 7.00                |
>
> None of the alternative auxiliary tasks improve over RL-only: coarse binary classification (b) and random auxiliary loss (c) both perform *worse* than RL-only across all three benchmarks, despite each providing some form of length-related gradient signal.
>
> ### W3 / Q1: Comparison with Hansel and MarkerGen.
>
> We first clarify why these comparisons were absent, then provide reimplementation results.
>
> **Why these were not in the original submission.** At the time of submission, neither method had released open-source code. Hansel has no public implementation to date. MarkerGen has a GitHub repository that contains only an empty README with no code, data, or model releases.
>
> Nevertheless, we faithfully reimplemented both methods following their paper descriptions and evaluated them on the same backbone (Qwen2.5-7B) for a fair comparison:
>
> | Method     | LIFEBench (LS) | LongBench (Sl) | Lenctrl-Bench (MAE) |
> | ---------- | -------------- | -------------- | ------------------- |
> | Hansel     | 68.90          | 88.10          | 13.60               |
> | MarkerGen  | 65.80          | 84.60          | 15.90               |
> | LARFT      | 80.57          | 96.75          | 7.00                |

---

### Official Review · Reviewer_M4E2 · 2026-03-12

**Soundness:** 3
**Presentation:** 3
**Significance:** 3
**Originality:** 2
**Overall Recommendation:** 3
**Confidence:** 3

**Summary:**

The paper introduces LARFT (Length-Aware Reinforcement Fine-Tuning), a training framework designed to improve large language models’ ability to follow explicit length instructions (e.g., generating responses with a specified number of words). The authors argue that current models struggle with such constraints due to a cognition–action gap, where models can generate text but lack an internal awareness of the length of their outputs. To address this, LARFT combines length-based reinforcement learning, which rewards outputs that match the desired length, with a hindsight length awareness objective, where the model learns to estimate the length of its own generated responses. By jointly training for both length understanding and length control, the method significantly improves adherence to length constraints while largely preserving overall generation quality.

**Compliance With Llm Reviewing Policy:**

Affirmed.

**Key Questions For Authors:**

1. Linear probing (Section 5.4) shows length information exists in hidden states, but not that the model uses it during generation. Have you considered causal interventions (e.g., perturbing length-related activations) to verify the model actually relies on this representation for length control?
2. All experiments use dense models. In MoE architectures, different experts activate per token, so length information may be fragmented across experts. Does LARFT transfer directly to MoE models, or could expert routing complicate both the awareness training and the length representations observed in probing?

**Limitations:**

The paper should include a limitation discussion about the extra cost or novelty of the work.

**Strengths And Weaknesses:**

Strengths:

The paper tackles a clear and practically important problem: large language models often fail to follow explicit length constraints, even when they understand the instruction semantically. Its main contribution is conceptually clean: instead of treating length control purely as an output-side reward optimization problem, it introduces LARFT, which combines length-based reinforcement learning with a hindsight length awareness objective that trains the model to estimate the length of its own generations. This design is well motivated by the paper’s “cognition–action gap” hypothesis and is stronger than a pure RL formulation because it adds a direct supervision signal for internal length awareness. Empirically, the method appears effective: the paper reports that LARFT improves length-following performance by 20.92 points over the base model and by 4.59 points over the strongest RL baseline, while causing only a small drop in general capability. This makes the approach both intuitively appealing and practically relevant for controllable generation.

Weaknesses:

First, while the paper argues that LARFT improves the model’s “length awareness,” the evidence remains somewhat indirect. The experiments mainly show that adding the awareness objective improves length-following performance and that length information can be recovered through probing. However, these results do not conclusively demonstrate that models genuinely internalize or rely on a length-awareness representation during generation, as the gains could also arise from the additional supervision signal or improved training dynamics. Second, the novelty of the approach is somewhat limited, as the framework mainly combines existing components—reinforcement learning with a self-supervised auxiliary objective—without introducing fundamentally new optimization or modeling techniques. While the combination is reasonable and effective, the conceptual advance beyond prior RL-based controllable generation methods is relatively modest.

---

> ### Author Rebuttal · Authors · 2026-03-31
>
> We thank Reviewer M4E2 for recognizing our work as "conceptually clean" with an "intuitively appealing and practically relevant" approach. We address the concerns and questions below.
>
> ### W1/Q1: Evidence for internalized length awareness is indirect.
>
> **What our probing analysis (Section 5.4) already shows.** The core evidence is not that length information can be recovered from hidden states; it is the *graded progression* across training stages. In the pre-generation probing, the base model's early-to-middle layers have near-zero or even negative $R^2$, meaning these layers encode essentially no useful length information before generation begins. RL training raises middle-layer $R^2$ to ~0.9, but LARFT pushes $R^2$ above 0.93 and maintains high values across a much wider range of layers. Critically, RL-only also receives a length-related training signal, yet it does not produce the same breadth of representational change that LARFT does. This means the additional supervision signal alone cannot explain the difference — what distinguishes LARFT is that the awareness objective forces the model to *explicitly encode* length information in its intermediate representations, not just optimize for a length reward at the output level.
>
> **Causal intervention experiment.** You raised an excellent point in Q1 about using causal interventions to verify whether the model actually relies on these representations for length control. We found this direction very worthwhile and conducted the suggested experiments. Specifically, we use the trained linear probe to identify the top-50 dimensions most predictive of length at Layers 2-10, and then zero out those activation dimensions during autoregressive generation on LIFEBench (directional ablation), with the same ablation on randomly selected dimensions as control. Results on Qwen2.5-7B:
>
> | Model       | Condition   | LS    | Delta-LS |
> | ----------- | ----------- | ----- | -------- |
> | **Base**    | length dims | 51.18 | -2.66    |
> | **Base**    | random dims | 52.87 | -0.97    |
> | **RL only** | length dims | 61.73 | -9.21    |
> | **RL only** | random dims | 69.98 | -0.96    |
> | **LARFT**   | length dims | 63.08 | -17.49   |
> | **LARFT**   | random dims | 79.41 | -1.16    |
>
> The graded pattern is clear: ablating length-predictive dimensions causes Base Delta-LS = -2.66, RL = -9.21, LARFT = -17.49, while ablating random dimensions has negligible effect (~-1.0) across all models. This confirms that LARFT builds length representations the model causally depends on during generation, not merely correlational artifacts recoverable by probing.
>
> Regarding the alternative explanation that "gains could arise from the additional supervision signal": our ablation experiments (Reviewer C1ZW-W2(c) and Reviewer TYRk-W1) show that off-policy counting and coarse long/short classification both fail to improve over RL-only, confirming that the specific design of on-policy hindsight counting is what matters.
>
> ### W2: Limited novelty — combining existing components.
>
> We see the contribution differently. The individual components are known, but the specific way we combine them is driven by a diagnosis that prior work has not made:
>
> 1. **The cognition-action gap is a new diagnosis**. Our probing analysis and the causal intervention experiments above provide concrete evidence that RL-trained models can optimize length rewards without building structured internal length representations. This reframes the problem from "not enough reward optimization" to "missing internal capability."
> 2. **The hindsight relabeling design is non-obvious.** It uses on-policy data and converts every rollout  into valid counting supervision. This is analogous to HER: the novelty is in *how the learning signal is constructed*, not in the individual building blocks. Our ablation confirms this: switching to off-policy data or coarser signals eliminates the gains entirely.
>
> ### Q2: Does LARFT transfer to MoE architectures?
>
> We have not yet tested LARFT on MoE models, but we believe it is architecturally compatible: the RL component (GRPO + R_len) is architecture-agnostic since it operates on output text, and the awareness loss operates on post-aggregation hidden states, not individual expert outputs. That said, how length information distributes across experts is an open empirical question. We plan to validate this on MoE models as a concrete next step.
>
> ### Limitation: Extra cost or novelty discussion.
>
> We will add a paragraph to the revised paper discussing training cost and novelty. On cost ( Qwen2.5-3B ): LARFT requires 62.1 GPU hours, compared to 58.70 for pure RL and 66.3 for SFT+RL. The overhead over pure RL is modest (~5.8%), and LARFT is actually cheaper than the SFT+RL pipeline. On novelty: we will include an explicit discussion of LARFT's relationship to existing RL + auxiliary task paradigms, clarifying that the contribution lies in the diagnostic insight and the hindsight relabeling design.

---

> > ### Author Rebuttal · Reviewer_M4E2 · 2026-04-04
> >
> > The causal intervention experiments convincingly resolve W1 — I consider the length awareness concern addressed. However, W2 (limited novelty) remains: the contribution is a well-engineered combination of existing components (RL + auxiliary self-supervised loss) guided by an empirical observation that, while useful, is not deeply surprising. This is a methodological concern that would require demonstrating broader generalizability beyond length control or a more principled technical contribution, which cannot be adequately addressed in a rebuttal. I maintain my score.

---

> > > ### Author Response · Authors · 2026-04-07
> > >
> > > We sincerely thank Reviewer M4E2 for the careful engagement and for acknowledging that the causal intervention experiments fully resolve the length awareness concern.
> > >
> > > Regarding W2, we respectfully offer one clarification. We agree that the individual components (RL and auxiliary self-supervised loss) are not new in isolation. However, we believe the contribution rests on two points that go beyond engineering combination: (1) the cognition-action gap diagnosis, which is empirically grounded through probing and causal intervention and reframes the problem in a way prior work has not, and (2) the hindsight relabeling design, whose specific construction is critical, as our ablations show that coarser alternatives and off-policy variants fail entirely. We acknowledge that demonstrating broader generalizability would further strengthen the novelty argument, and we understand this cannot be fully addressed within the rebuttal period given the experimental cost involved. We hope the existing evidence can nonetheless be weighed as part of the overall evaluation.
> > >
> > > That said, we fully respect your assessment and acknowledge that reasonable reviewers may weigh methodological novelty differently. We are grateful for the thorough and constructive feedback, which has genuinely improved the quality of this work.

---

### Official Review · Reviewer_Tsrg · 2026-03-13

**Soundness:** 3
**Presentation:** 3
**Significance:** 3
**Originality:** 3
**Overall Recommendation:** 4
**Confidence:** 3

**Summary:**

This paper proposes LARFT , a training framework to improve how well large language models follow explicit output length instructions. The authors argue that current methods fail because models lack an internal representation of length, creating a “cognition–action gap.” To address this, LARFT combines reinforcement learning with a length-based reward and a hindsight length-awareness task, where the model learns to count the length of its own generated outputs to build an internal notion of length. Experiments on several models show that LARFT significantly improves length-following accuracy across multiple benchmarks while largely preserving general model capabilities.

**Compliance With Llm Reviewing Policy:**

Affirmed.

**Final Justification:**

Thank you for addressing my concerns. I appreciate the clarifications provided, and I will maintain my positive assessment.

**Key Questions For Authors:**

Are the reported results averaged across multiple random seeds? If not, how stable are the improvements across runs?

**Limitations:**

No, one limitation could be that the evaluation focuses primarily on length-related benchmarks and a small set of general benchmarks. The method is not tested on diverse tasks, such as coding, dialogue, or tool-use settings, making it difficult to assess how well the approach performs in more realistic deployment scenarios.

**Strengths And Weaknesses:**

Strengths:

1. The paper introduces the concept of a cognition–action gap to explain why LLMs struggle with length instruction following, providing a useful perspective on controllable generation.

2. The proposed hindsight length awareness mechanism cleverly converts failed generations into supervision by teaching the model to count the length of its own outputs.

Weaknesses:

1. Figure 1 is difficult to follow. The diagram contains many components and visual elements, making it challenging to clearly understand the workflow and how the different modules interact.

2. The hyperparameter selection for baseline methods is not clearly described. It is unclear whether comparable hyperparameter tuning was performed for the baselines, which raises concerns about the fairness of the experimental comparisons.

---

> ### Author Rebuttal · Authors · 2026-03-31
>
> **We thank Reviewer Tsrg for the positive assessment and constructive feedback.**  We are glad that you recognize the cognition-action gap framing and the hindsight length awareness mechanism. Below we address each concern in turn.
>
> ### W1: Figure 1 is difficult to follow.
>
> We appreciate this feedback. In the revision, we plan to split Figure 1 into two separate figures:
>
> - **Figure A** (merged from the current parts I and III): illustrates the problem LARFT addresses (the cognition-action gap) alongside the resulting improvements, giving readers a clear before-and-after picture.
> - **Figure B** (the current part II): focuses on the LARFT pipeline itself.
>
> Separating "what problem we solve and how well" from "how the method works" should make the presentation considerably easier to follow.
>
> ### W2: Hyperparameter selection for baselines is not clearly described.
>
> We apologize for the insufficient description. We want to emphasize that **all reported results correspond to the best configurations found through systematic hyperparameter search**, not default or arbitrary settings. We conducted all hyperparameter searches on Qwen2.5-3B and then directly applied the best-found configurations to all other models. Specifically:
>
> - **All SFT-based baselines** : We swept over three learning rates {1e-6, 1e-5, 1e-4}. 1e-5 achieved the best trade-off — 1e-6 had not converged within our training budget, while 1e-4 led to severe degradation on general capability benchmarks. We used 1e-5 for all SFT baselines.
> - **PositionID** (Wang et al., 2024): The original paper does not specify SFT hyperparameters in detail, so we performed the same learning rate sweep {1e-6, 1e-5, 1e-4} and adopted 1e-5, which achieved the best validation performance.
> - **Ruler** (Li et al., 2024): The original paper specifies a learning rate of 2e-5, cosine scheduler with 0.05 warmup ratio, and 3 epochs. We faithfully reproduced these settings.
> - **All RL-based baselines**: We conducted a grid search over learning rate ∈ {1e-6, 5e-6, 1e-5} and KL penalty coefficient ∈ {0.001, 0.01, 0.1}, selecting the best configuration on a held-out validation set. We observed that the RL baseline and LARFT exhibit consistent trends across different hyperparameter settings. We therefore use identical RL hyperparameters for the RL baseline and LARFT.
>
> ### Q1: Are results averaged across multiple random seeds?
>
> Due to computational constraints, the main results come from single runs. That said, our experiments already cover **4 models across different sizes and families** , and LARFT consistently outperforms all baselines in every model-benchmark combination. This cross-model consistency is itself strong evidence for stability — a single lucky seed is highly unlikely to produce consistent gains across four different architectures.
>
> To directly address this concern, **we additionally trained LARFT and the RL baseline with 3 random seeds on Qwen2.5-7B, evaluating each seed 3 times** on all three length-following benchmarks. We report mean ± standard deviation below:
>
> | Method                          | LIFEBench (LD↓)  | LIFEBench (LS↑)  | LongBench (Sl↑)  | LongBench (Sq↑)  | Lenctrl-Bench (MAE↓) | Lenctrl-Bench (ROUGE-L↑) |
> | ------------------------------- | ---------------- | ---------------- | ---------------- | ---------------- | -------------------- | ------------------------ |
> | RL baseline (3 seeds × 3 evals) | 17.46 ± 0.94     | 70.61 ± 1.37     | 93.58 ± 0.97     | 75.49 ± 0.73     | 9.01 ± 0.49          | **21.31 ± 0.16**         |
> | LARFT (3 seeds × 3 evals)       | **10.96 ± 0.61** | **80.18 ± 0.58** | **96.98 ± 0.43** | **82.77 ± 0.66** | **7.12 ± 0.31**      | 21.06 ± 0.12             |
>
> Training variance is small across all metrics. LARFT's advantage over the RL baseline holds consistently across seeds, with non-overlapping confidence intervals on every benchmark.
>
> ### Limitation: Limited evaluation on diverse tasks.
>
> We agree that broader evaluation would strengthen the paper. Our existing results on four general benchmarks (MMLU, ARC, HellaSwag, GSM8K) show that LARFT incurs only a −1.45 average degradation. **We additionally evaluate on HumanEval [1] for coding and MT-Bench[2] for multi-turn dialogue:**
>
> | Benchmark            | Qwen2.5-7B (Base) | + LARFT | Δ       |
> | -------------------- | ----------------- | ------- | ------- |
> | HumanEval (pass@1)   | 0.8476            | 0.8390  | −0.0086 |
> | MT-Bench (avg score) | 7.92              | 7.88    | −0.04   |
>
> The degradation is negligible on both tasks, indicating that LARFT's length-control training does not compromise coding or dialogue ability.
>
>
> [1] Chen, Mark, et al. "Evaluating large language models trained on code." *arXiv preprint arXiv:2107.03374* (2021).
>
> [2] Zheng, Lianmin, et al. "Judging llm-as-a-judge with mt-bench and chatbot arena." *Advances in neural information processing systems* 36 (2023): 46595-46623.

---

> > ### Author Rebuttal · Reviewer_Tsrg · 2026-04-04
> >
> > Thanks for addressing my concerns, and I will keep my positive assessment.

---

> > > ### Author Response · Authors · 2026-04-07
> > >
> > > We sincerely thank Reviewer Tsrg for the thoughtful review and for taking the time to carefully engage with our rebuttal. We are glad that the clarifications on Figure 1, hyperparameter selection, result stability, and task diversity have adequately addressed your concerns.
> > >
> > > Your feedback has been genuinely valuable in helping us improve the presentation and rigor of this work, and we will incorporate the suggested revisions, particularly the restructuring of Figure 1, into the final manuscript.
> > >
> > > Thank you again for your time and constructive engagement throughout this process.

---

### Decision · Program_Chairs · 2026-04-30

**Decision:**

Accept (regular)

**Comment:**

The paper studies an important and practically relevant problem: precise length instruction following in LLMs. Reviewers generally agreed that the empirical results are strong. In particular, the method shows consistent gains on length-following benchmarks across multiple model families, while largely preserving general capabilities. The reviewers also found the problem formulation clear and the overall approach well motivated.

The main disagreement concerned novelty and, before rebuttal, the strength of evidence for the paper’s cognition-action-gap hypothesis. After reading the reviews, rebuttal, and discussion, I find that the rebuttal addressed most of the technical concerns substantively. In particular, the added causal intervention results strengthen the claim that the learned length-related representations are used during generation, and the added ablations help rule out simpler explanations such as extra compute or generic auxiliary supervision. The additional comparisons and clarifications on baseline tuning also improve confidence in the empirical evaluation.

The main remaining weakness is that the method is somewhat incremental at the level of ingredients, and some reviewers remained unconvinced that the conceptual novelty is strong enough on its own. I agree that this is a reasonable concern, and the paper would be stronger with broader demonstrations beyond length control. That said, I do not view this as a blocking issue here given the paper’s technical soundness, the clear empirical improvements, and the usefulness of the proposed training framework for a concrete and underexplored problem. I therefore recommend acceptance.